*Report*

# STAT3 sustains tumorigenicity following mutant KRAS ablation

Stephen D'Amico [ID][1,3], Varvara Kirillov[1], Jingxuan Liu[2,4], Zhijuan Qiu[1], Xinyuan Lei [ID][1], Hong Qin[1], Brian S Sheridan[1] & Nancy C Reich [ID][1✉]

## Abstract

**Oncogenic KRAS mutations underlie some of the deadliest human cancers. Genetic or pharmacological KRAS inactivation produces mixed outcomes and frequent relapse. Mechanisms of tumor resistance to KRAS inhibition remain poorly understood. We present evidence that STAT3 supports tumor growth following KRAS depletion. Using a conceptual framework of pancreatic ductal adenocarcinoma, we show that cancer cells that survive CRISPR-mediated ablation of mutant KRAS are dependent on STAT3 function to maintain tumorigenicity. Mechanistically, the combined loss of mutant KRAS and STAT3 disrupts a core transcriptional program of cancer cells critical to oncogenic competence. This in turn impairs tumor growth in mice and enhances immune rejection, leading to tumor clearance. We propose that the STAT3 transcriptional program operating in cancer cells enforces their malignant identity, rather than providing classical features of transformation, and shapes cancer persistence following KRAS inactivation. Our findings establish STAT3 as a critical enforcer of oncogenic identity in KRAS-ablated tumors, revealing a key vulnerability.**

**Keywords** KRAS; STAT3; Oncogene Dependence; Pancreatic Cancer
**Subject Categories** Cancer; Immunology; Signal Transduction

## Introduction

Pancreatic ductal adenocarcinoma (PDAC) is a highly aggressive malignancy characterized by rapid progression and exceptional resistance to anticancer treatment (Halbrook et al, 2023). Despite KRAS mutations driving 90% of PDAC cases, direct KRAS inhibition has had limited success, in part due to tumor heterogeneity and adaptive resistance (Briere et al, 2021; Liu et al, 2024; Punekar et al, 2022). Historically, therapies have assumed that KRAS-driven cancers are entirely dependent on KRAS signaling. However, the degree of oncogenic KRAS addiction can vary across individual tumor cells,

challenging this notion (Muzumdar et al, 2017; Punekar et al, 2022; Salmon et al, 2023; Singh et al, 2009; Yuan et al, 2018). Results suggest that even the most potent inhibition of oncogenic KRAS alone may not be sufficient for curing cancer in humans.

Pancreatic tumorigenesis is a process that involves the gradual accumulation of mutations in oncogenes (e.g., KRAS GTPase) and tumor suppressor genes (e.g., TP53, CDKN2A, and SMAD4)(Hayashi et al, 2021). Cooperative mutations of oncogenes and tumor suppressors appear to function within a specific cellular context that designates tissue oncogenic competence (Baggiolini et al, 2021; Hsieh et al, 2024; Weeden et al, 2023). The signal transducer and activator of transcription 3 (STAT3) has been shown to contribute to early stages of inflammation in pancreatic intraepithelial neoplasia, activated in part through IL-6/gp130 receptor cytokines (Corcoran et al, 2011; Fukuda et al, 2011; Lesina et al, 2011). Classical activation of STAT3 occurs following tyrosine 705 phosphorylation by receptor-associated Janus kinases (JAKs) or other tyrosine kinases (Huynh et al, 2019; Philips et al, 2022). Tyrosine phosphorylation promotes STAT3 dimerization and its DNA-binding ability, although STAT3 has also been shown to facilitate gene expression without tyrosine phosphorylation. STAT3 does not display oncogenic activity in vitro, and it is not required for mutant KRAS to transform cells (D'Amico et al, 2024). However, PDAC cells genetically depleted for STAT3 or for mutant KRAS shared partially overlapping transcriptional signatures. For this reason, we investigated the hypothesis that following depletion of oncogenic KRAS, STAT3 could maintain tumorigenicity initiated by KRAS. Our findings indicate that complete inhibition of advanced-stage PDAC is contingent on ablation of mutant KRAS and the concomitant inactivation of STAT3. STAT3 is a key regulator sustaining malignancy following mutant KRAS inactivation (Philips et al, 2022).

## Results and discussion

### STAT3 correlates with KRAS independence in human PDAC

Studies with epithelial cancer cells expressing mutant KRAS have shown that at an advanced tumor stage, cancer cells can have a reduced dependency on the KRAS oncogene (Lim and Counter, 2005; Muzumdar et al, 2017; Singh et al, 2009; Yuan et al, 2018).

---

[1]Department of Microbiology and Immunology, Stony Brook University, Stony Brook, NY 11794, USA. [2]Department of Pathology, Stony Brook University, Stony Brook, NY 11794, USA. [3]Present address: Stony Brook Medicine Division of Nephrology & Hypertension, Stony Brook, NY, USA. [4]Present address: St. Luke's University Health Network, St. Louis, MO, USA. ✉E-mail: nancy.reich@stonybrook.edu

 

Specific gene expression signatures have been derived for cancer cells reliant on KRAS, designated KRAS-dependent (KRAS_sig/KRAS-type), or less reliant on KRAS, designated KRAS-independent (RSK_sig/RSK-type) (Singh and Settleman, 2009; Yuan et al, 2018). To examine the potential association between STAT3 expression and the degree of oncogenic KRAS dependency, we analyzed gene expression profiles in datasets of human PDAC tumor samples. Tumors were classified by the sum of mRNA expression values (z-scores) of individual genes that comprise KRAS-dependent or KRAS-independent gene signatures. Two clear trends emerged. First, tumor samples stratified into either KRAS-dependent or KRAS-independent subtypes regardless of cancer stage or treatment history (Figs. 1A and EV1A,B)(Singh et al, 2009; Yuan et al, 2018). Second, STAT3 mRNA and STAT3-regulated gene expression correlated with the tumor samples classified as KRAS-independent (Figs. 1A and EV1A,C,D) (Dauer et al, 2005). We evaluated PDAC tumor specimens from more than 20 patients with primary and metastatic PDAC for activation of ERK mitogen-activated protein kinase. Classically, ERK activity is a downstream readout of KRAS activation and identifies KRAS-dependent cancer cells (Klomp et al, 2024). Immunohistochemistry for activated ERK showed extensive heterogeneity, consistent with the evolution of KRAS independence (Fig. EV1E,F).

## Loss of STAT3 prevents tumor formation after KRAS ablation

To evaluate the contribution of STAT3 to the tumorigenicity of PDAC cells, we used CRISPR-mediated gene editing to eliminate endogenous KRAS and/or STAT3 in the human PANC-1 cell line. Although PANC-1 cells bear an activated KRAS G12D mutation, they have been classified as KRAS-independent type (Singh and Settleman, 2009). We first isolated independent KRAS KO and STAT3 KO clones, confirmed loss of KRAS and STAT3 by protein expression, and tested their ability to form orthotopic tumors in nude mice (Figs. 1B,C and EV1G). Both the STAT3 KO and the KRAS KO cells formed tumors, but tumor latency increased with KRAS ablation. To evaluate the dependency of the KRAS KO PANC-1 cells on STAT3, we depleted STAT3 in KRAS KO tumorigenic cells and generated KRAS/STAT3 double knockout (DKO) clones. While STAT3 loss did not alter PANC-1 cell proliferation in vitro (Fig. EV1H), its absence in KRAS-ablated cells prevented orthotopic tumor formation in vivo (Fig. 1C). We also tested the ability of these cells to develop tumors as subcutaneous xenografts (Fig. 1D). KRAS KO PANC-1 cells retained the ability to form subcutaneous tumors in nude mice, but more slowly than parental PANC-1 controls, consistent with published observations (Muzumdar et al, 2017). However, only one of the three DKO PANC-1 clones was able to form any tumors. The results suggest that under conditions of KRAS depletion, inactivation of STAT3 can completely abolish pancreatic tumor growth.

To gain a better understanding of the roles of STAT3 in tumor maintenance, we generated a panel of CRISPR-edited murine KPC cell lines (KRAS$^{G12D/+}$; p53$^{R172H/+}$) (Hingorani et al, 2005), each mimicking complete inactivation of the KRAS and/or STAT3 genes (D'Amico et al, 2024; D'Amico et al, 2018; Ischenko et al, 2021). Three types of isogenic cell lines were examined: KRAS KO, STAT3 KO, and KRAS/STAT3 DKO. DNA sequencing revealed insertion-deletion mutations within the targeted STAT3 and KRAS alleles,

and Western blot analysis showed no detectable STAT3 or KRAS$^{G12D}$ protein expression (Figs. 1E and EV1I). Loss of STAT3 alone minimally affected the growth of cells in culture or as tumors in nude mice (Figs. 1F and EV1J) (D'Amico et al, 2024; D'Amico et al, 2018). KRAS KO clones produced tumors in nude mice when implanted subcutaneously or orthotopically into the pancreas, although with a longer latency than parental controls (Ischenko et al, 2021; Muzumdar et al, 2017). Thus, ablation of KRAS alone did not eliminate their tumorigenic properties in nude mice. In contrast, the tumorigenicity of DKO cell lines was either attenuated or abolished (i.e., no tumor formation after 60 days of observation) (Figs. 1F and EV1K). We used limiting dilution assays in nude mice to estimate the frequency of tumor-initiating cells (TICs) (Hu and Smyth, 2009). The TICs ranged from ~0.7% in parental KPC cells to ~0.034% in DKO1 cells (~20-fold change), requiring more than $2 \times 10^4$ implanted DKO cells for tumor formation (Fig. 1G).

We next determined the impact of an intact immune system on pancreatic tumor growth. Tumor formation was assessed following orthotopic implantation of isogenic KPC cell lines into the pancreas of wild-type C57Bl/6 mice. KRAS KO cells were severely impaired in tumor-forming capacity, but their tumorigenicity was not eliminated completely, as shown previously (Fig. 1H,I)(Ischenko et al, 2021). However, co-deletion of KRAS and STAT3 (i.e., DKO) fully abrogated tumor growth in wild-type recipient mice. Histology showed that pancreatic tumors derived from KPC parental cells resembled well-differentiated PDAC with prominent glandular structures, whereas STAT3 KO and KRAS KO tumors exhibited a mesenchymal-like morphology as reported previously (Fig. 1J) (D'Amico et al, 2024). Importantly, there was no evidence of malignancy in the pancreata of wild-type mice transplanted with DKO cell lines after 90 days of observation. The data indicate that STAT3 is needed to sustain pancreatic tumor growth under conditions of intact immunity following the genetic inactivation of mutant KRAS.

Genetic depletion of mutant KRAS from PDAC cancer cells has been shown to greatly reduce but not abolish their sustained tumorigenicity (Ischenko et al, 2021). Here, we show that KRAS-depleted cells are reliant on the function of STAT3 to form tumors in immunocompetent mice. Even human PANC-1 cells that contain a dozen driver mutations (cBioPortal; Cancer Cell Line Encyclopedia), depend on the function of STAT3 to form tumors following KRAS ablation.

## Combined loss of KRAS and STAT3 disrupts a core transcriptional program in PDAC cells

To gain insight into the mechanisms by which STAT3 mediates continued tumor growth following KRAS inactivation, we performed RNA sequencing analysis of KPC cells and their STAT3 KO, KRAS KO, and DKO derivatives. Using fold change ≥4 and false discovery rate <0.05 cut-offs, we identified 2000 genes as differentially expressed by DKO vs. parental cells. Pairwise differential expression analyses are displayed as comparative scatter plots (Fig. 2A). We compared RNA-seq results of parental and knockout cells with modules of twenty genes corresponding to signatures involved in stem cell maintenance, epithelial identity, and cell adhesion (GSEA/MSigDB). Pathway enrichment analyses revealed expression of these gene sets was reduced in the DKO samples (Figs. 2B and EV2A,B) although they maintained

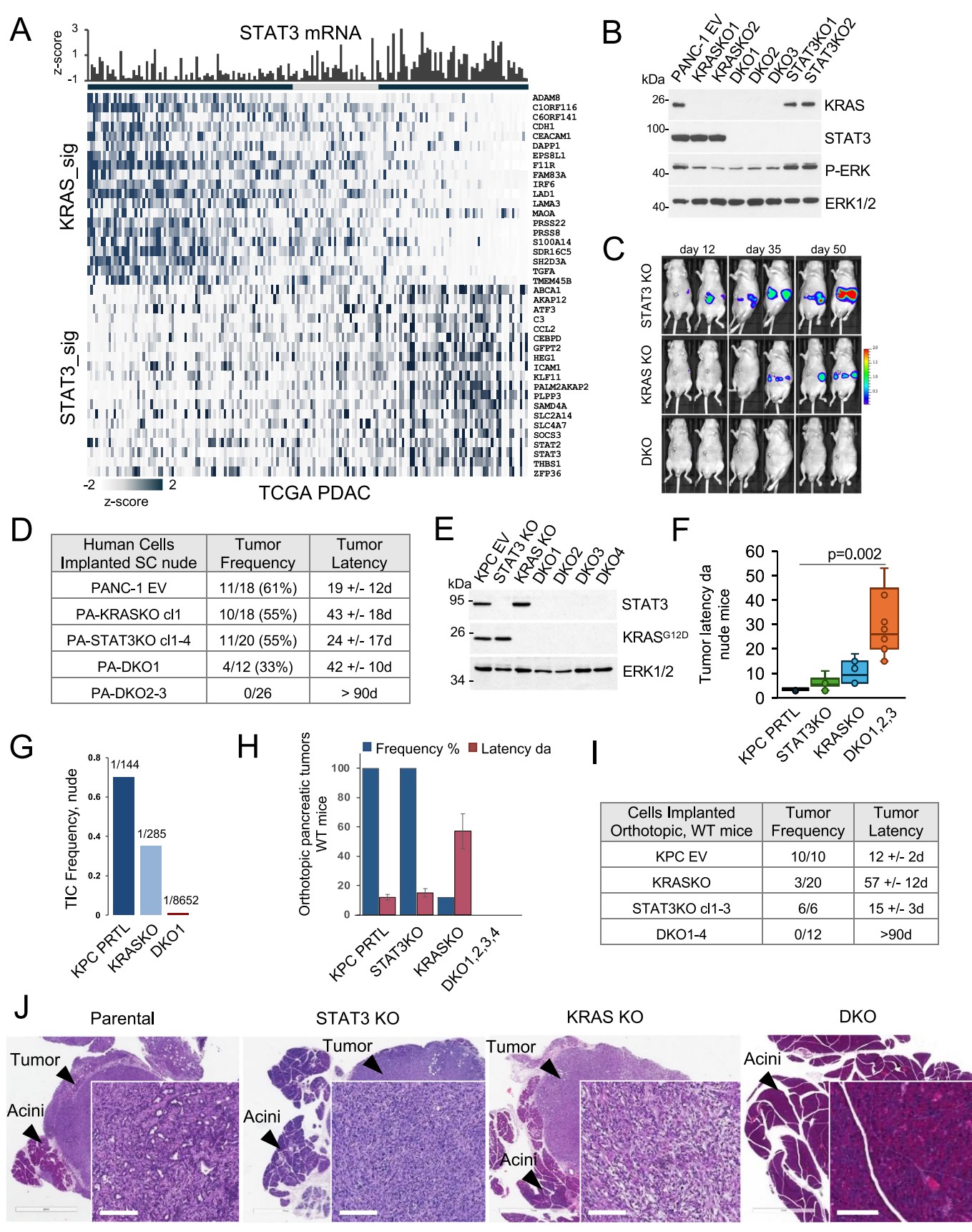

WT mice

◀ **Figure 1. STAT3 maintains survival of mutant KRAS-ablated pancreatic tumors.**

(A) Human PDAC expression profiles from The Cancer Genome Atlas (TCGA) were downloaded as z-scores relative to diploid samples from cBioPortal (http://www.cbioportal.org) along with additional tumor and clinical annotations. Gene expression profiles of tumor samples ($n = 150$) were segregated according to KRAS dependency (KRAS_sig) (Singh and Settleman, 2009) and STAT3-dependent gene regulation (Dauer et al, 2005). Top panel displays the corresponding STAT3 mRNA expression (z-scores) of each tumor. (B) Western blot of human PANC-1 parental cells expressing empty lentiviral vector (EV) and cells with CRISPR-mediated knockout of endogenous KRAS$^{G12D}$ (KRAS KO, two clones), STAT3 (STAT3 KO, two clones), or KRAS$^{G12D}$ and STAT3 (DKO, three clones). Antibodies to KRAS, STAT3, ERK1/2 and phospho-ERK were used. (C) Representative in vivo bioluminescence imaging of nude mice following orthotopic implantation of STAT3 KO cells, KRAS$^{G12D}$ KO cells, or DKO cells derived from the human PANC-1 cell line transduced with the luciferase gene. Days denote time following implantation. (D) Tumor formation of human PANC-1 empty vector (EV) cells and derived independent KRAS KO, STAT3 KO, and DKO clones (cl). Tumor frequency and latency (days, ~1 mm) are noted. (E) Western blot of KRAS$^{G12D/+}$ p53$^{R172H/+}$ KPC cells with parental empty vector (EV) and derived isogenic clones showing ablation of endogenous KRAS$^{G12D}$ (KRAS KO), STAT3 (STAT3 KO), and four independent clones with a double ablation of KRAS$^{G12D}$ and STAT3 (DKO) with antibodies to STAT3 and KRAS. ERK1/2 is a loading control. (F) Boxplots showing subcutaneous tumor latencies (days, ~1 mm) in nude mice of parental KPC EV cells or isogenic knockout derivatives noted ($n = 5$ biological replicates). Boxplots show center line as median, box limits as upper and lower quartiles, and whiskers the 1.5 interquartile range. Significance determined using one-way ANOVA and Tukey's post-hoc test and shown for DKO and parental cells. (G) Limiting dilution assays in nude mice were used to calculate the frequency of tumor-initiating cells (TICs) of parental KPC, KRAS KO, and DKO1 cells. (H) Parental KPC cells or knockout derivative cells were orthotopically implanted into the pancreas of syngeneic wild-type mice (C57BL/6). Tumor formation frequency (%) and latency in days are shown on the same axis ($n = 6$–20 biological replicates). Frequency and latency are not shown for the DKO cells since there was no tumor development. Significance determined using one-way ANOVA and Tukey's post-hoc test. (I) Results of tumor development in wild-type mice following orthotopic implantation into the pancreas with KPC empty vector, STAT3 KO, KRAS KO, or DKO cells shown in (H). Tumor frequency shown as the number of mice with tumors/total number of mice. Tumor latency (~350 mm³) shown in days (d). (J) Representative histological images of H&E staining of KPC pancreatic tumors following orthotopic implants of designated cells in wild-type mice. Inset scale bar 200 mm. Source data are available online for this figure.

significant levels of phosphorylated MEK1/2 and ERK1/2 (Fig. EV2C). The most salient feature underlying the changes in DKO cells was the perturbation of transcription factors (at least 60 genes) controlling lineage specificity, differentiation, proliferation, and survival (Figs. 2C,D and EV2D). These were examined based on a predictive list of core transcription factors that regulate tumor cell fates (Reddy et al, 2021). Analyses showed that many deregulated transcription factors were those involved in differentiation and lineage specification (e.g., ELF3, FOXA1, FOXA2, HNF4A, KLF4, KLF5, SNAI, etc.), and that DKO cells underwent reprogramming to a far greater extent than either STAT3 or KRAS single knockouts. Western blot analyses confirmed the reduced expression of HNF4A, FOXA1, FOXA2, KLF4, GATA5 in DKO cells (Figs. 2E and EV2D). These findings indicate that STAT3 works to sustain a core transcriptome profile within KRAS-ablated PDAC tumor cells.

We ectopically re-expressed STAT3, KRAS$^{G12D}$ and select transcription factors in the DKO cells to determine whether reconstitution could reverse cell morphology and recover tumorigenic capacity. Expression of STAT3 or mutant KRAS by lentiviral transduction in DKO4 cells partially restored tumorigenicity in nude mice, and their pairwise expression further increased the frequency of tumor formation (Figs. 2F and EV3A). The STAT3Y705F allelic mutation was severely compromised in recovering tumorigenicity, indicating that STAT3 tyrosine phosphorylation and DNA binding are major contributors to tumorigenicity. Among the transcription factors tested, expression of FOXA1 and FOXA2 achieved the highest recovery of tumorigenicity. Co-expression of FOXA1 and FOXA2 further increased the frequency of tumors and accelerated tumor growth (Figs. 2F and EV3B). In contrast, no reversal was observed following ectopic expression of HNF1A, HNF4A, KLF5, or MYC. These data demonstrate that the combined loss of KRAS and STAT3 (or their downstream targets FOXA1/2) significantly impacts tumorigenic growth in vivo. Ectopic expression of FOXA1 or FOXA2 also generated the highest degree of phenotypic conversion, generating tumors that were similar morphologically to parental KPC tumors with increased epithelial differentiation (Fig. EV3C).

Gene expression in DKO cells transduced with *Foxa1* or *Foxa2* was evaluated by RNA-seq and compared with parental KPC or DKO cells. Analysis showed that ectopic expression of FOXA1 and FOXA2 caused DKO cells to revert to a state that resembled parental KPC cells in terms of stem cell maintenance, cell adhesion, and epithelial differentiation, supporting their respective roles in the maintenance of the KRAS mutant phenotype (Fig. 2G). The DKO cells supplemented with FOXA1/2 also exhibited a gene signature more closely associated with PDAC ductal subtype 2, distinguishing a more malignant cell state (Fig. EV3D)(Cui Zhou et al, 2022; Peng et al, 2019).

To ascertain whether transcriptome alterations of FOXA1 and FOXA2 reconstituted cell lines could be projected onto human PDAC, we computed gene expression signatures derived from the top 100 up- or downregulated genes. PDAC tumor samples from TCGA were segregated according to RAS dependency signature and differentiation status. Alignment revealed that, similar to STAT3, FOXA1/2 regulated expression signatures co-segregated with epithelial (EPI) KRAS-dependent (RDI) tumors (Pearson's $r > 0.6$) and were inversely correlated with mesenchymal (MES) KRAS-independent (RSK) tumors (Pearson's $r > 0.6$)(Fig. EV2I). RNA-seq analysis of human PANC-1 cells and their DKO derivatives identified more than 2600 differentially expressed genes in the DKO cells at a threshold of fourfold difference or greater. Among the top downregulated cellular pathways in DKO cells were oxidative phosphorylation, ribosome biogenesis, and cytoplasmic translation, whereas the upregulated pathways included axon guidance and muscle tissue development (Fig. EV4A). In addition, a unique subset of transcription factors that regulate various biological processes were again differentially expressed in the DKO cells (Fig. EV4B). Together, the data demonstrate that combined loss of KRAS and STAT3 (or their downstream targets) leads to cellular reprogramming that significantly impacts tumorigenic growth.

STAT3 can regulate genes directly or indirectly, and databases with STAT3 chromatin immunoprecipitation DNA sequencing (ChIP-seq) information have identified *Foxa1* as a STAT3-specific target (https://maayanlab.cloud/Harmonizome/gene_set/STAT3/

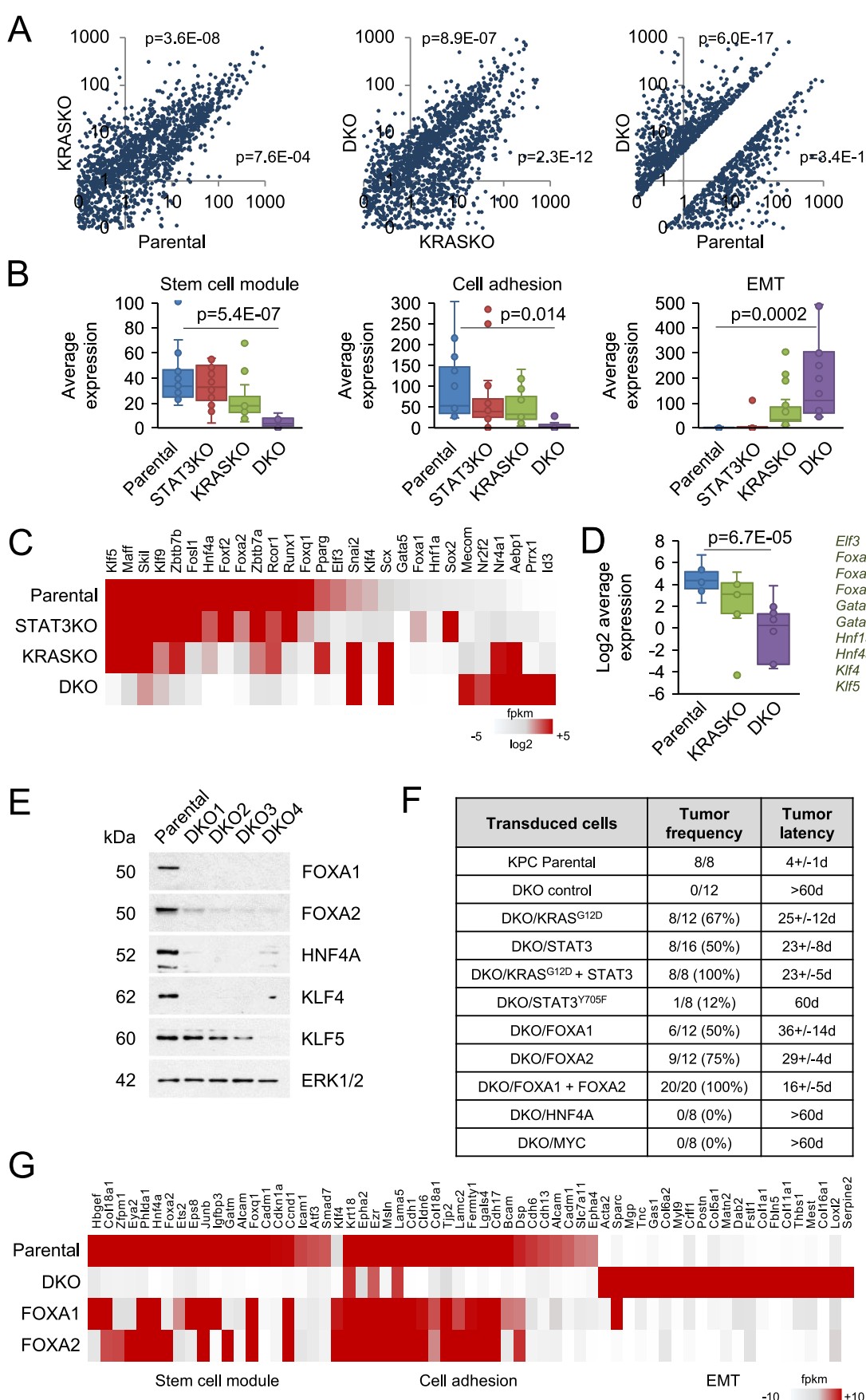

**Figure 2. Transcriptional profiling of cells with combined loss of KRAS and STAT3.**

(A) Differential expression of genes displayed as scatter plots for KPC EV parental, KRAS KO and DKO cells derived from RNA-seq data with >4-fold difference and FDR < 0.05. (B) Pathway scoring using the top 20 genes expressed in KPC EV parental cells and derived KRAS KO, STAT3 KO, and DKO cells that correspond to stem cell maintenance, cell adhesion, and EMT signatures from The Molecular Signatures Database (MSigDB). Boxplots show center line as median, box limits as upper and lower quartiles, and whiskers the 1.5 interquartile range. Significance determined using one-way ANOVA and Tukey's post-hoc test. (C) Comparative heatmaps of RNA-seq expression data of transcription factors in KPC EV parental and derived KRAS KO, STAT3 KO, and DKO4 cells. (D) Summary boxplots of differential expression for a subset of transcription factors shown that regulate lineage specificity from RNA-seq data of KPC parental, KRAS KO and DKO cells. Boxplots show center line as median, box limits as upper and lower quartiles, and whiskers the 1.5 interquartile range. Significance determined using one-way ANOVA and Tukey's post-hoc test. (E) Western blot showing the protein expression of a subset of transcription factors in KPC EV parental cells and four double KO (DKO) clones. ERK1/2 is a loading control. (F) Restoration of tumorigenicity in nude mice of DKO4 cells following transduction with mutant *Kras^{G12D}*, *Stat3*, *Stat3^{Y705F}*, *Hnf4a*, *Myc*, *Foxa1*, *Foxa2* or the indicated combinations compared with parental KPC cells. Tumor frequency (tumors/implantation site) and latency ~1 mm size (days) are provided. (G) Heatmaps of RNA-seq expression data of KPC parental cells, DKO4 cells, and DKO4 cells reconstituted with *Foxa1* or *Foxa2* overexpression. Specific gene signatures were obtained from MSigDB as in Fig. 3B. Source data are available online for this figure.

ENCODE+Transcription+Factor+Targets) (Diamant et al, 2025). We investigated whether there was enrichment of STAT3 target genes in KRAS KO cells that may contribute to their sustained tumorigenicity. We performed a stringent ChIP-seq with antibodies to tyrosine-phosphorylated STAT3 or control antibodies in parental KPC cells, KRAS KO and DKO cells. The results identified both previously reported and novel STAT3 gene targets, including *Stat3* itself, *Fos, Jak3, Syt12, Rasa3*, and *Twf1* that are associated with proliferation, invasion, and immune modulation (Fig. EV4C) (Johansen et al, 2023; Liu et al, 2020; Seaton et al, 2024). The increased representation of these gene targets in the KRAS KO cells may reflect the ability of STAT3 to support KRAS independence. Our assays did not identify *Foxa1* or *Foxa2*, possibly signifying their indirect regulation by STAT3.

Prior studies of PDAC tumor cells depleted for KRAS uncovered transcriptome alterations in key regulatory networks involved in cell fate determination, self-renewal, and differentiation (He et al, 2018; Kaestner, 2010; Kinisu et al, 2021; Novak et al, 2020; Takahashi and Yamanaka, 2006). STAT3 is not a direct downstream effector of KRAS; however, gene expression changes in KRAS KO and STAT3 KO cells have shown a partial overlap in KRAS and STAT3-dependent genes (D'Amico et al, 2024). Here we find that ablation of both mutant KRAS and STAT3 (DKO) disrupts a core program of transcription factors that includes the pioneer factors FOXA1 and FOXA2. FOXA1/2 are essential for differentiation of endodermal-derived tissues and pancreas development, and they have been reported to contribute to the PDAC malignant phenotype (Gao et al, 2008; Geusz et al, 2021; Kaestner, 2010; Milan et al, 2019; Roe et al, 2017; Song et al, 2010). Most importantly, we were able to partially restore the tumorigenicity of DKO cells with transduction of mutant *Kras, Stat3, Foxa1* and *Foxa2*. The results demonstrate plasticity in the tumorigenic properties of cancer cells and that loss of KRAS and STAT3 (or their downstream targets) significantly impacts tumorigenic growth. The central finding supports the existence of an embedded transcriptional program of self-renewal that likely evolves with oncogenic KRAS and is maintained by STAT3 following the loss of KRAS.

## STAT3 depletion enhances tumor rejection

We investigated whether depletion of STAT3 in the tumor cells could improve the efficacy of pharmacological KRAS inhibition. Since activation of STAT3 has been reported in response to tumor regression

by RAS/RAF/MAPK inhibitors (Lee et al, 2014; Salmon et al, 2023), we first evaluated STAT3 activation and T-cell infiltration of parental KPC tumors from mice following treatment with a protocol of KRAS pathway inhibition. KRAS^{G12D} inhibitor monotherapy or immune checkpoint inhibitor therapy alone has limited effects on PDAC tumor regression (Li et al, 2023; Punekar et al, 2022). However, we have shown that treatment with a combination of KRAS^{G12D} and MEK inhibitors (MRTX1133 and GSK1120212) with T-cell activators (PD1, CTLA4, and CD40 antibodies) regresses PDAC tumors in mice by more than 60% within two weeks (Li et al, 2023). To evaluate STAT3 activation during regression, we treated mice bearing orthotopic KPC tumors with a combination of MRTX1333, GSK1120212, and PD1, CTLA4, and CD40 antibodies or with control IgG antibodies intraperitoneally. Immunohistochemistry was performed on the KPC tumors to measure STAT3 activation with antibodies to tyrosine 705 phosphorylated STAT3, and T-cell immune infiltration with antibodies to CD8a. We found that both STAT3 tyrosine phosphorylation and CD8 T-cell infiltration were increased in tumors from mice following treatment with the KRAS pathway inhibitor cocktail (Li et al, 2018) (Fig. 3A). The elevated activation of STAT3 may reflect a survival response of tumor cells to KRAS inhibition.

We next investigated whether inactivation of STAT3 could enhance tumor regression. Orthotopic tumors formed by parental KPC cells expressing empty vector or STAT3 KO cells were measured at the onset of the experiment (day 1). Mice bearing these tumors received four treatments with the KRAS inhibitory cocktail during a seven-day period and tumors were evaluated by weight on the seventh day (day7T) (Fig. 3B). Within one week, STAT3 KO tumors showed a significant reduction in tumor weight compared to mice with control KPC tumors, including complete regression (Fig. 3C). The results provide preclinical evidence that STAT3 depletion can enhance the antitumor effects of KRAS combination therapy.

Clinical targeting of STAT3 remains to be actualized, although some small molecules, oligonucleotides, peptides, and PROTACs are in early phase 1 trials (Wang et al, 2024). For this reason, we used genetic depletion of STAT3 to demonstrate its contribution to tumorigenicity following pharmacological inhibition or genetic ablation of KRAS. Given the challenge of direct STAT3 inhibition, alternative approaches such as targeting STAT3-regulated genes warrant exploration. Inhibitors of Janus kinases have recently been reported to improve the efficacy of some anticancer immunotherapies (Mathew et al, 2024; Zak et al, 2024). However, JAK inhibitors do not affect the regression of KPC tumors. This is not unexpected since JAK inhibitors attenuate immune cell function.

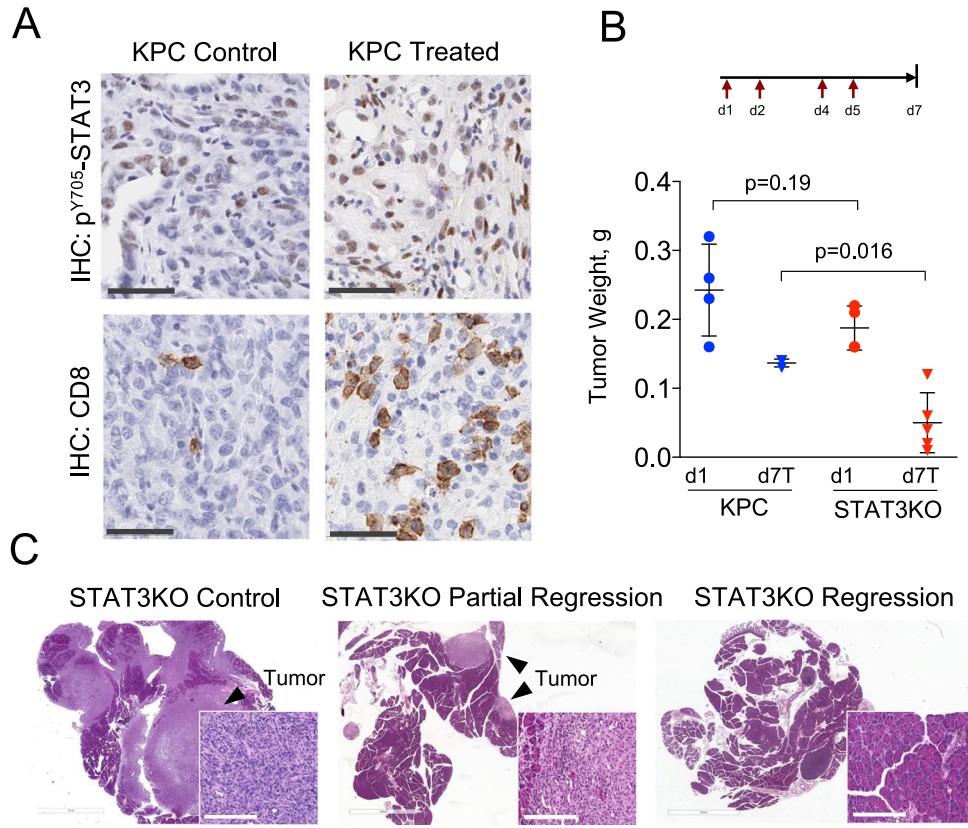

**Figure 3. Tumor regression following combination therapy.**

(A) Representative immunohistochemistry (IHC) of orthotopic tumors formed by KPC parental cells (Li et al, 2018) in wild-type mice treated with control IgG antibodies or mice treated by administration of a cocktail of MRTX1133, GSK1120212, and antibodies to PDCD1, CTLA4, and CD40 as previously described (Li et al, 2023). Tumors were stained by IHC with antibodies to phosphotyrosine 705 STAT3 (pY705-STAT3) or CD8a (CD8). Insert scale bar 100 mm. (B) Effect of combination drug treatment on the growth of orthotopic tumors formed by KPC EV cells or isogenic STAT3 KO cells. Top bar indicates timeline of intraperitoneal injections (red arrow) in days. Tumor weights are shown in strip plots for individual mice at the onset of the experiment day 1 (d1) and for mice harvested on day 7 following drug administration on days 1, 2, 4, and 5 (d7T). Each data point represents one mouse, $n = 4$–5 per group. Significance determined using one-way ANOVA and Tukey's post-hoc test. Plots show median center line and upper and lower quartile lines. Untreated mice bearing tumors from KPC or STAT3 KO cells by day 7 had tumor weights equal to or greater than 0.5 g (not shown). (C) Representative histological H&E images of tumors formed by STAT3 KO cells in control mice or mice following drug administration at day 7 from (B). Inset scale bar 200 mm. Source data are available online for this figure.

## Loss of KRAS and STAT3 in PDAC cells modulates tumor immunity

Since PDAC cells depleted for both mutant KRAS and STAT3 (DKO) do not produce tumors in wild-type mice and therefore cannot be analyzed, we investigated an independent model system with a doxycycline-inducible KRAS$^{G12D}$ and mutant p53$^{R172H}$ (termed iKRAS) (Collins et al, 2012). STAT3 expression was also consequential in this iKRAS model, even though transient expression of mutant KRAS does not reflect tumor evolution with reduced dependency on KRAS. We used CRISPR-mediated gene editing to isolate several independent iKRAS clones depleted for STAT3 expression (Fig. 4A). Both iKRAS and derived STAT3 KO cell clones formed tumors in nude mice with administration of doxycycline. However, when transplanted into the pancreas of syngeneic wild-type mice, STAT3 KO iKRAS cell clones exhibited reduced and varied tumor-forming frequency (Fig. EV5A). Flow cytometry analyses of tumors formed by STAT3 KO cells showed an increase in T-cell number with a higher CD8 to CD4 ratio relative to control tumors expressing empty vector (Fig. 4B). Since

withdrawal of doxycycline extinguishes expression of the KRAS transgene, we assessed short-term regression of tumors (Collins et al, 2012). We found that the median weight of STAT3 KO tumors was reduced compared to iKRAS controls as early as four days post-doxycycline withdrawal (Fig. 4C). In vivo imaging with luciferase-expressing cells showed some of the STAT3 KO tumors regressed to undetectable levels by seven days following doxycycline withdrawal (Fig. EV5B). Overall, the data indicate that co-deletion of STAT3 and KRAS facilitates tumor rejection in wild-type recipients.

To extend these observations, we performed single-cell RNA sequencing (scRNAseq) of pooled iKRAS control or STAT3 KO orthotopic tumors from wild-type mice treated with doxycycline. High-dimensional single-cell sequencing data were embedded using two-dimensional Uniform Manifold Approximation and Projection (UMAP). Unsupervised density clustering segregated cells corresponding to populations of tumor cells, tumor associated macrophages (TAMs), cancer-associated fibroblasts (CAFs), granulocytes/neutrophils, T cells, B cells, endothelial cells, and normal pancreatic cells (Fig. 4D). Based on captured viable cell populations

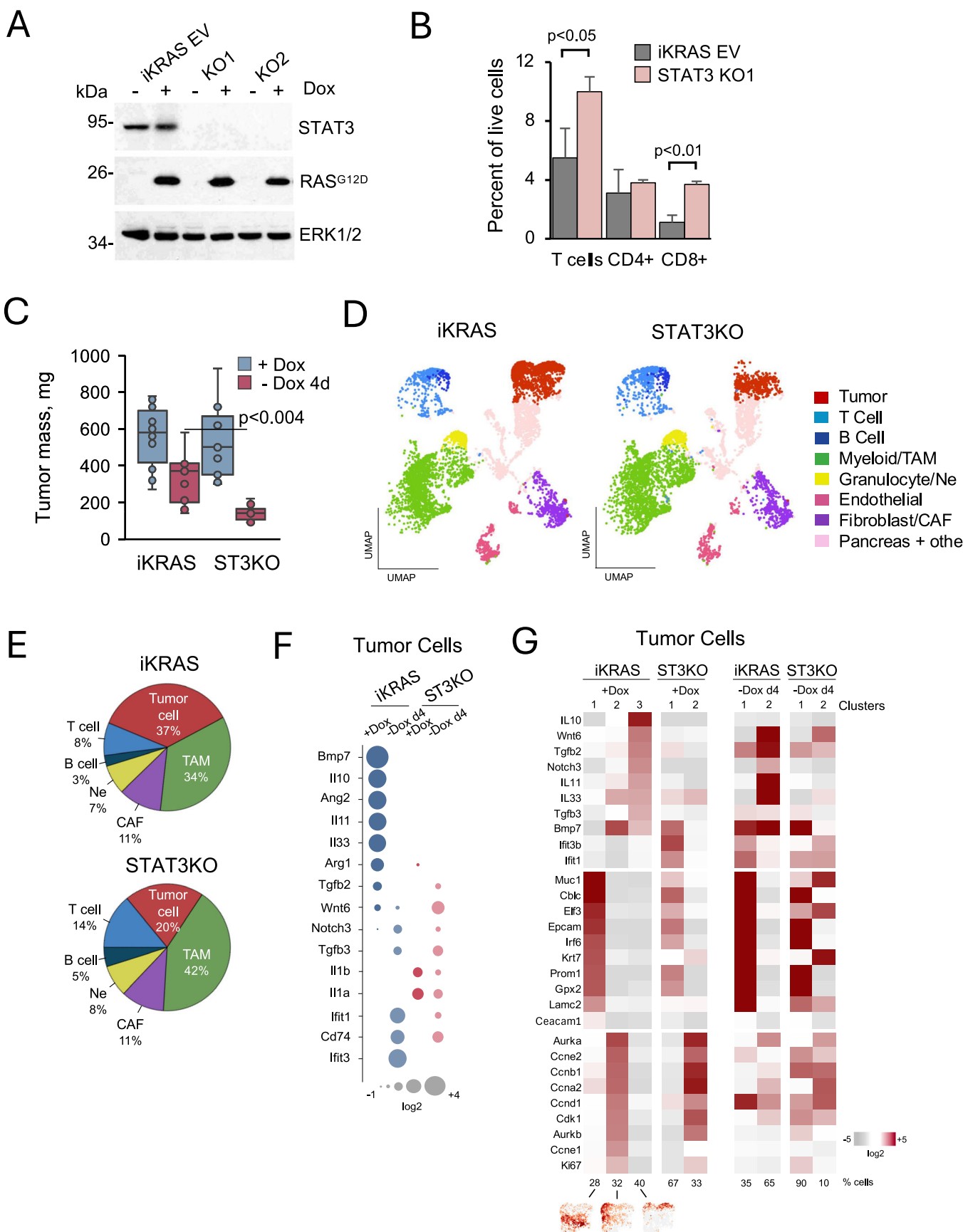

◄ **Figure 4. Influence of STAT3 loss in an inducible mutant KRAS PDAC model.**

(A) Western blot of iKRAS PDAC parental cells expressing empty vector (EV) and two independent STAT3 KO derivatives, untreated (−) or treated (+) with doxycycline. Protein expression was detected with antibodies to STAT3 and KRAS. Antibodies to ERK1/2 were used as loading controls. (B) Flow cytometry of tumors formed by orthotopic implantation of parental iKRAS cells or derived STAT3 KO cells into the pancreas of wild-type FVB mice. Mice were administered doxycycline, and pancreatic tumors that formed were pooled from 2–3 individual mice. Single cells were dispersed and stained with fluorescently tagged anti-TCRb, anti-CD3e, anti-CD4, or anti-CD8a. Results of two biological replicates with triplicate technical replicates. Data are presented as mean ± SD, two-tailed $t$ test. (C) Pancreatic tumor weights (mg) are shown following orthotopic implantation of iKRAS EV parental or STAT3 KO cells into wild-type FVB mice. Mice were either treated with doxycycline (+Dox) or treated and withdrawn from doxycycline for 4 days (-Dox 4 d) ($n > 6$). Boxplots show center line as median, box limits as upper and lower quartiles, and whiskers the 1.5 interquartile range. Significance determined using one-way ANOVA and Tukey's post-hoc test and shown following doxycycline removal of between iKRAS and STAT3 KO cells (-Dox 4 d). (D) UMAP projection images derived with Loupe browser software from scRNA-seq of pancreatic tumors based on captured viable cell populations. Analyses were performed following orthotopic implantation of iKRAS EV parental cells (left) or STAT3 KO cells (right) into the pancreas of wild-type FVB mice followed by treatment of mice with doxycycline. Tumors were pooled from 2 to 3 individual mice. Main cell types are displayed in clusters distinguished by color based on multiple canonical marker genes. Some of these included *Onecut2* (tumor cell), *Trac* (T cell), *Cd79* (B cell), *Fap* (fibroblast), *Adgre1* (myeloid/macrophage), *Csf3r* (granulocyte/neutrophil), and *Cd93* (endothelial cells). (E) Pie charts display the relative percent of malignant tumor cells, TAMs, CAFs, Neutrophils (Ne), B cells and T cells in captured viable cell proportions of the pancreatic tumors formed by iKRAS or STAT3 KO cells in FVB mice treated with doxycycline derived from scRNA-seq in (D). (F) Dot heatmap expression comparison of four scRNA-seq aggregated datasets from captured viable cell proportions of the pancreatic tumors. Expression of a subset of immunoregulatory genes in cancer cells from orthotopic tumors formed by iKRAS EV control cells or derived STAT3 KO cells in wild-type FVB mice. Mice were treated with doxycycline (+Dox) or treated and withdrawn from doxycycline for 4 days (-Dox 4 d). (G) Heatmaps of differential gene expression from scRNA-seq individual datasets of captured viable tumor cell proportions from iKRAS EV or STAT3 KO orthotopic tumors formed in mice treated with doxycycline (+Dox) or treated and then withdrawn from doxycycline for 4 days (-Dox 4 d). The tumor cells within each tumor type segregated into distinct Clusters consistent with gene signatures for KRAS dependency, cell cycle, or immunomodulation. Numbers below correspond to percent of tumor cell population designated to a specific cluster. UMAP projection images are shown below for three marker genes in tumor cells of iKRAS +Dox clusters 1–3. Source data are available online for this figure.

and proportions, tumors derived from doxycycline-treated mice bearing STAT3 KO cells were estimated to contain a relatively lower percentage of tumor cells (20% vs 37%) and a higher percentage of immune cells than tumors from control iKRAS cells (Fig. 4E). Differential gene expression was performed against scRNAseq datasets of cancer cells in tumors from mice treated with doxycycline or withdrawn from the drug for four days. Evaluation of a subset of immunomodulatory genes showed higher transcript levels in the iKRAS tumor cells relative to STAT3 KO cells, and this was most apparent prior to the removal of doxycycline (e.g., IL-10, Arg1) (Fig. 4F). Although both iKRAS and STAT3 KO cells formed tumors when mutant KRAS was turned on, there was an overall trend of higher immune suppression in the iKRAS cancer cells. Analysis of infiltrating TAMs and CD8 T cells in the tumors showed modest differences between tumor types. TAMs in the iKRAS tumors showed a reduction in pro-inflammatory gene expression (M1 polarization) and an increase in genes associated with alternative activation (M2 polarization) compared with STAT3 KO tumors (Fig. EV5C) (Boutilier and Elsawa, 2021; Sica and Mantovani, 2012). The CD8 T-cell population of iKRAS tumors exhibited higher expression of some markers indicative of dysfunction/exhaustion (e.g.,*Cd160*, *Lag3*, *Gmzb*) compared to STAT3 KO tumors, and lower expression of markers for activation (e.g., *Il2*, *Bcl2*) (Fig. EV5D) (Andreatta et al, 2021; Kallies et al, 2020). Overall, the data indicate that STAT3 activity contributes to the ability of PDAC tumor cells to evade immunity.

Activation of STAT3 is mediated by various tyrosine kinases, most notably by Janus kinases (JAKs) stimulated in response to the family of IL-6 cytokines. Studies have shown that following KRAS and MEK drug inhibition of pancreatic tumor cells, there is an increase in STAT3 tyrosine phosphorylation, and this increase can be ablated by a JAK inhibitor (Miyazaki et al, 2025). We previously found that cancer-associated fibroblasts in tumors formed by KRAS KO cancer cells have an increased production of IL-6 in comparison to KRAS-expressing tumors. In this report, we found the depletion of mutant KRAS in the KPC tumor cells increases their expression of IL-6, IL-11 and the IL-11

receptor. Therefore, following KRAS depletion, it is conceivable that increased STAT3 tyrosine phosphorylation is due to IL-6 cytokine family activation (Fig. 3A).

Intratumor heterogeneity of PDAC tumors has been considered a challenge to successful therapeutic intervention (Bailey et al, 2016; Collisson et al, 2011; Moffitt et al, 2015; Peng et al, 2019; Verbeke, 2016). To investigate heterogeneity in the iKRAS and STAT3 KO tumor cell populations, we interrogated the scRNAseq data for gene signatures that correspond to KRAS dependency, cell cycle, and cytokine production (Fig. 4G). Intratumor heterogeneity within the tumor cell population was clearly evident and distinguished by differing patterns of gene expression. The iKRAS expressing tumor cells from doxycycline-treated mice displayed three main clusters: Cluster 1, enriched for expression of murine KRAS dependency genes (e.g., *Muc1*, *Elf3*, *Epcam*) (D'Amico et al, 2024; Ischenko et al, 2021); Cluster 2, enriched for cell cycle genes (e.g., *Aurka*, *Ccnb1*, *Cdk1*); and Cluster 3, enriched for immune modulatory cytokines (e.g., *IL-10*, *Wnt6*, *Tgfb2*). Although KRAS signaling is often correlated with proliferation, the analyses indicate that the KRAS and cell cycle signatures are expressed primarily in distinct tumor cell populations. In addition, the cytokine modulatory cluster is separate from KRAS dependency and cell cycle clusters. The finding that gene signatures in iKRAS tumor cells identify distinct cancer cell clusters highlights the need for combination therapies that individually target mutant KRAS, cell proliferation, and immune suppression for successful intervention. The cancer cell population from the STAT3 KO tumors conspicuously lacked the immunomodulatory cytokine cluster 3, providing further evidence that STAT3 contributes to immune suppression as well as the degree of tumor heterogeneity. Our findings identify a specific role for STAT3 in maintaining tumorigenic properties. Rather than promoting initial stages of cell transformation, we reveal a role of STAT3 in sustaining tumor cell identity.

The limited clinical success of KRAS inhibitors as monotherapies points to the need to identify multiple means by which advanced-stage cancers maintain stemness and immune evasion

following depletion of oncogenic KRAS. In this report, we used genetic ablation of mutant KRAS as a tool to mimic its pharmacological inactivation and to uncover PDAC tumor vulnerabilities. We provide evidence for an underlying role of STAT3 in maintaining oncogenic identity and immune suppression following loss of mutant KRAS in cancer cells. The results have important implications for successful therapeutic intervention.

# Methods

### Reagents and tools table

| Reagent/resource | Reference or source | Identifier or catalog number |
|---|---|---|
| **Experimental models** | | |
| Adults male and female mice (8–10 weeks age) | | |
| C57Bl6/6J | Jackson Laboratory | Strain #:000664 |
| NU/J | Jackson Laboratory | Strain #:002019 |
| FVB/NJ | Jackson Laboratory | Strain #001800 |
| KPC cells | Hingorani et al, 2005 | |
| iKRAS cells | Collins et al, 2012 | |
| HEK293T cells | ATCC | |
| Phoenix E cells | ATCC | |
| PANCi1 cells | ATCC | |
| **Recombinant DNA** | | |
| LentiCRISPRv2 puro | Addgene | |
| pBluescript KS (+) lentiviral IRES GFP vector lentiCRISPRv2 puro | Addgene D'Amico et al, 2018, Addgene | |
| **Antibodies** | | |
| P-ERK1/2 | Cell Signaling | 9101 |
| E-cadherin | Cell Signaling | 3195 |
| c-FOS | Cell Signaling | 2250 |
| FOXA2 | Cell Signaling | 8186 |
| KLF4 | Cell Signaling | 4038 |
| KLF5 | Cell Signaling | 51586 |
| P-MEK1/2 S217/221 | Cell Signaling | 9154 |
| MYC | Cell Signaling | 5605 |
| NFkBp65 | Cell Signaling | 8242 |
| KRASG12D | Cell Signaling | 14429 |
| aSMA | Cell Signaling | 19245 |
| STAT3 | Cell Signaling | 9139 |
| P-STAT3 Y705 | Cell Signaling | 9145 |
| KRAS | Santa Cruz | Sc-30 |
| HNF4 | ABclonal | A11496 |
| GATA5 | AssayBioTech | R12-2154 |
| ERK1/2 | EMD Millipore | 05-157 |
| FOXA1 | Abcam | Ab170933 |
| P-Y705 STAT3 | R&D systems | CST9145 |
| CD8a | R&Dsystems | CST98941 |
| Control rabbit IgG | Cell Signaling | 2729 |
| P-Y705 STAT3 | Cell Signaling | D3A7 |

| Reagent/resource | Reference or source | Identifier or catalog number |
|---|---|---|
| **Oligonucleotides and other sequence-based reagents** | | |
| Murine sgKRAS RNA (5'-gtggttggagctgatggcgt-3') | This study | |
| Murine sgSTAT3 RNA (5'-gcagctggacacacgctacc-3' and 5'-gtacagcgacagcttcccca-3') Human sgKRAS RNA (5'-gtagttggagctgatggcgt-3') Human sgSTAT3 RNA (5'-tgtacagcaccggccgatg -3'). | This study Mou et al, 2017 This study | |
| **Chemicals, enzymes, and other reagents** | | |
| DMEM and FBS | Gibco | |
| Doxycycline | Sigma | BE0146 |
| Puromycin | Sigma | BE0164 |
| Hygromycin B | Sigma | BE0016 |
| TranssIT-LT1 | Mirus | 9005 |
| Collagenase/Hyaluronidase | Stem Cell Technology | |
| Liberase-DL | Roche | |
| DNAseI | Sigma | |
| Wizard Genomic DNA Purification Kit | Promega | |
| NE-PER extraction reagents | ThermoFisher | |
| PureLinkRNA | ThermoFisher | |
| RNAlater | Corning | |
| Opti-MEM | Syd Lab Inc | |
| D-luciferin | Chemietek | |
| MRTX1133 | Selleck Chem Inc | |
| GSK1120212 | Bio X Cell | |
| Anti-PD1 | Bio X Cell | |
| Anti-CTLA4 | Bio X Cell | |
| Anti-CD40 | Cell Signaling | |
| SimpleChIP Plus | | |
| **Software** | | |
| Loupe Browser | 10X Genomics | |
| Gene Set Enrichment ELDA software | Broad Institute (http://bioinf.wehi.edu.au/software/elda/ http://bioinf.wehi.edu.au/software/elda/) | |
| **Other** | | |

## Mammalian cells and reagents

Murine KRAS[G12D] p53[R172H] (KPC) cells (Hingorani et al, 2005; Li et al, 2018) and inducible KRAS[G12D] p53[R172H] (iKRAS) pancreatic epithelial cells (A9993) (Collins et al, 2012) were gifts. Human PANC-1, Phoenix-E and HEK293T cells were obtained from ATCC. All cells were maintained in DMEM supplemented with 5% FBS and 1× antibiotic/antimycotic (Gibco). For proliferation assays, cells were grown for 96 h, counted with Vi-CELL XR Analyzer (Beckman Coulter), and doubling times were calculated. Cell viability was measured by trypan blue staining. KRAS transgene expression in iKRAS cells was induced by incubating with 1 μg/ml doxycycline hyclate (Sigma) for 24 h. Stable cell lines were generated and isolated by FACS isolation for GFP or selection in the appropriate antibiotic, puromycin (2 ug/ml) or hygromycin B (100 ug/ml) followed by single clone isolation using glass cloning cylinders.

## Lentiviruses and plasmids

For isogenic CRISPR/Cas9-mediated murine knockout cells, we used murine sgKRAS RNA (5′-gtggttggagctgatggcgt-3′) and sgSTAT3 RNA (5′-gcagctggacacacgctacc-3′ or 5′-gtacagcga-cagcttcccca-3′) cloned into LentiCRISPRv2 puro (Addgene) (Mou et al, 2017; Sanjana et al, 2014). Human knockout cells were generated using sgKRAS RNA (5′-gtagttggagctgatggcgt-3′) (Muzumdar et al, 2017) and sgSTAT3 RNA (5′-tgtacagcaccggcc-gatg-3′). Genomic DNA from putative knockouts was isolated using the Wizard Genomic DNA Purification Kit (Promega), PCR-amplified, cloned into pBluescript KS (+) (Addgene) and confirmed by Sanger sequencing. Primer sequences are available upon request. Retroviral plasmids encoding murine *Foxa1, Foxa2, Gata5, Hnf1a, Hnf4a, Kras$^{G12D}$, c-Myc* and luciferase were obtained from Addgene. The lentiviral IRES GFP vector (adapted from the pWPXL/pEF1a backbone) encoding murine STAT3 alleles was used as previously described (D'Amico et al, 2018). Phoenix-E and HEK293T cells were transiently transfected with the TransIT-LT1 (Mirus) transfection reagent according to the manufacturer's instructions. Production and collection of recombinant retroviruses were done according to standard protocols.

## Histology and immunohistochemistry

Tissues were excised, immersion fixed in ≥5 volumes of 4% paraformaldehyde for 48 h, transferred to 70% ethanol, and processed by the Stony Brook University Research Histology Core. Paraffin-embedded formalin-fixed 5-μm sections were stained with hematoxylin and eosin (H&E) for histology. Human pancreatic adenocarcinoma tumor samples were obtained as paraffin-embedded tissue specimens from the Stony Brook Medicine BioBank and prepared by the Histology Core. Immunohistochemistry was performed by iHisto (Salem, MA) and Histowiz (Brooklyn, NY). H-scores for human tumors and phosphorylated ERK (CST D13.14.4E) represent the sum of malignant cell staining intensity in the tumors (negative 0, weak 1, moderate 2 or strong 3) multiplied by the percentage of stained cells. Slides were scored independently by two investigators. Comparative immunohisto-chemistry staining for phosphorylated tyrosine 705 STAT3 (CST9145) and CD8a (CST98941) was performed in KPC tumors from four mice in two biological replicate experiments by visual scoring above background of four random microscopic fields.

## Expression analyses

Western blotting was performed using whole cell extracts prepared by lysing cells in buffer containing 10 mM Tris-HCl, pH 7.4, 150 mM NaCl, 1 mM EDTA, 10% glycerol, 1% Triton X100, 40 mM NaVO4, 0.1% SDS, and 1× protease inhibitors (Roche). Nuclear and cytoplasmic fractions were prepared using NE-PER nuclear and cytoplasmic extraction reagents (ThermoFisher). Proteins were detected with antibodies to P-ERK1/2 (9101), E-cadherin (3195), c-FOS (2250), FOXA2 (8186), KLF4 (4038), KLF5 (51586), P-MEK1/2 S217/221 (9154), MYC (5605), NFkB p65 (8242), RASG12D (14429), α-SMA (19245), STAT3 (9139), P-STAT3 Y705 (9145) (Cell Signaling); KRAS (sc-30, Santa Cruz); HNF4A (A11496, ABclonal); GATA5 (R12-2154, AssayBioTech); ERK1/2 (05-157, EMD Millipore); FOXA1 (ab170933, Abcam). Total

cellular RNA was isolated using PureLink RNA (ThermoFisher) according to the manufacturer's specifications and phenol-chloroform extracted. Pancreatic tumor tissue was incubated overnight at 4 °C in ≥5 volumes of RNA*later* solution (Thermo-Fisher). Whole transcriptome RNA-sequencing (RNA-seq) with bioinformatics was performed by Novogene Corp. (http://en.novogene.com). Gene ontology (GO) classifications were generated by Novogene or the online GO Enrichment Analysis tool (http://geneontology.org).

Human pancreatic adenocarcinoma clinical data were down-loaded from cBioPortal (http://www.cbioportal.org) or from the Amsterdam UMC (AUMC) database (Dijk et al). Human tumor samples were classified as KRAS-dependent/KRAS-type, or KRAS-independent/RSK-type by calculating the sum of individual mRNA expression values (z-scores) of genes that were previously characterized as KRAS-dependent or independent and are listed in the figures (Singh et al, 2009; Yuan et al, 2018). A murine KRAS dependency signature was used to calculate RAS dependency scores and was previously described (D'Amico et al, 2024; Ischenko et al, 2021). STAT3 signature scores were computed from a set of regulated genes listed in the figure (Dauer et al, 2005). Additional pathway activity scores were determined using curated hallmark gene sets from the Molecular Signatures Database (www.gsea-msigdb.org) based on the average fpkm values (stem module M1999; hallmark EMT M5930, and cell adhesion M9500). Gene Set enrichment analysis was performed using the application available from the Broad Institute (http://software.broadinstitute.org/gsea/. Scatter plots and heatmaps were generated using online Heatmap-per software or Microsoft Excel.

## Tumorigenicity in mice

All animal studies complied with ethical regulations for animal testing and research approved by the Institutional Animal Care and Use Committee of Stony Brook University (protocol 2011-0356). We used adult (8–10 weeks old) male and female mice from The Jackson Laboratory, strains C57BL/6 J, FVB/NJ, and NU/J. Subcutaneous implantations were performed using $10^4$ murine or $4 \times 10^5$ human cells in 100 μl of Matrigel diluted 1:10 with Opti-MEM (Corning). Tumor latency was defined as the period between implantation of tumorigenic cells and the appearance of tumors ~1 mm. Orthotopic injections into the pancreas were performed with $10^4$ cells in diluted Matrigel using standard procedures (Kim et al, 2009). The animals were observed for tumor development by palpation or bioluminescence by two independent investigators. Tumor latency was defined as period between implantation of cells and tumor growth to ~350 mm³. The frequency of tumor-initiating cells was calculated using online ELDA software (http://bioinf.wehi.edu.au/software/elda/), and the number of cells implanted subcutaneously ranged from $10^2$ to $10^4$ (Hu and Smyth, 2009). For iKRAS tumor formation, the drinking water was provided ad libitum and supplemented with 2% sucrose, doxycy-cline hyclate (0.5 mg/ml), neomycin (0.5 mg/mL) and ampicillin (1 mg/mL). Water was changed every 3–4 days. For biolumines-cence imaging, mice were injected intraperitoneally with 200 μl 15 mg/ml D-luciferin (Syd Lab Inc.) and imaged using the Lumina III in vivo imaging system (IVIS) with Living Image Software (PerkinElmer). Drug treatment cocktails were administered by IP injection and were composed of anti-PD1 (BE0146), anti-CTLA4

(BE0164), and anti-CD40 (BE0016) antibodies from Bio X Cell at 1 mg/kg in PBS, and with MRTX1133 (Chemietek) at 10 mg/kg and GSK1120212 (Selleck Chem lnc) at 2 mg/kg in 5% DMSO, 40% PEG300, 5% Tween-80, and 45% saline. For flow cytometry, tumor cells were dissociated with 1× collagenase/hyaluronidase (Stem Cell Technologies). Cells were incubated with unconjugated anti-CD16/CD32 prior to staining with the following fluorophore-conjugated antibodies from BioLegend, Invitrogen, or BD Biosciences: anti-CD45 (APC-eF780), anti-TCRβ (PerCp/Cy5.5), anti-CD8α (BV786), anti-CD4 (BV605), anti-CD3e (BV421), anti-TCRd (PE), anti-CD11b (APC), anti-F4/80 (PE/Cy7) and anti-Ly6G (PE/Dazzle594). Live/dead dye was purchased from Invitrogen. Cells were stained protected from light at 4 °C for 20 min. After staining, cells were fixed with 2% paraformaldehyde (Electron Microscopy Sciences) for 20 min prior to data acquisition on a LSRFortessa (BD) or an Aurora (Cytek). Data were analyzed with FlowJo software (Tree Star). The data represent three experiments with tumors pooled from 2 to 3 mice.

## ChIP analyses

Chromatin immunoprecipitation was performed with Cell Signaling Technology SimpleChIP Plus enzymatic chromatin IP kit with magnetic beads (#9005). Briefly, formaldehyde treatment of tissue culture cells was used to crosslink proteins to DNA. Nuclei were prepared, chromatin was digested with micrococcal nuclease, and sonication was used to lyse nuclei. Cross-linked chromatin was immunoprecipitated overnight with antibodies and reagents from Cell Signaling Technology: control rabbit IgG (2729) or phospho-STAT3 antibody (Tyr705)(D3A7). Immunocomplexes were collected with ChIP-grade protein G magnetic beads (#9006), stringently washed, eluted, and reverse cross-linked. DNA was purified, and DNA libraries were prepared with NEBNext Ultra II DNA library prep kit for Illumina (NEB#E7645S) for next-generation sequencing (NG) by Novogene Corp. (NovaSeq PE 150). Bioinformatics of genome mapping, peak calling, peak annotation, and differential analysis was performed with BWA, MACS2, PeakAnnotaotr_Cpp, and diffbind software by Novogene.

## scRNA-Seq

Orthotopic tumors formed in FVB male mice with iKRAS parental control (A9993) (Collins et al, 2012) or iKRAS STAT3 KO cells were pooled from 2 to 3 individual mice at day 0 (~300–400 mg) or following doxycycline removal for 4 days (day 4) ( ~ 100–300 mg). Single cell suspensions were prepared by finely mincing tumor tissue with a razor blade followed by incubation with 1× collagenase/hyaluronidase (StemCell Technology), 0.5 mg/ml Liberase-DL (Roche), and 0.1 mg/ml DNase I (Sigma) in DMEM for 45 min at 37 °C with continuous mixing. Single-cell RNA sequencing and analysis were performed by the Cold Spring Harbor Laboratory Single-Cell Biology Shared Resource, Genomics Technology Development. Cells were loaded into the 10x Genomics microfluidics device along with 10X Genomics gel beads (kit v2 PN-120237) containing barcoded oligonucleotides, reverse transcription (RT) reagents, and oil, resulting in gel beads in emulsion. The scRNA-seq library preparation followed the manufacturer's protocol (10x Genomics) using the Chromium Single Cell 3-Library. Libraries were paired-end sequenced using the

Illumina HiSeq 4000 sequencing system. Genomics Cell Ranger pipeline (version 3.0.1) was used to convert single-cell sequencing files and secondary analysis was with Loupe Cell Browser v3.0.1 (10X Genomics).

## Statistics and reproducibility

Statistical analysis was performed using two-tailed Student's $t$ test, one-way ANOVA and Tukey's post-hoc test as appropriate for the dataset. An FDR-adjusted $P$ value ($q$ value) was calculated for multiple comparison correction. Individual mice, tumors, and tumor cell lines were considered biological replicates. Technical replicates were in triplicate. Statistical details for each experiment are noted in the corresponding figures and figure legends. The micrographs (H&E and IHC images) represent two or more biological replicate experiments. For the quantification of IHC, cells were counted manually with an average of 10–100 fields per tumor. All data are presented as mean ± SD.

## Data availability

Data was deposited in DRYAD as curated Microsoft Excel files for murine KPC cell RNA-seq data, and human PANC-1 RNA-seq data https://datadryad.org/dataset/doi:10.5061/dryad.1vhhmgqzc, https://doi.org/10.5061/dryad.5hqbzkhct, and https://datadryad.org/dataset/doi:10.5061/dryad.p5hqbzkxs. The scRNA-seq data have been deposited in the GEO database under accession code GSE275858.

The source data of this paper are collected in the following database record: biostudies:S-SCDT-10_1038-S44319-025-00563-w.

## Peer review information

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

## Acknowledgements

This work was supported by NIH grant RO1CA236389, the Carol M. Baldwin Breast Cancer Research Award, and the Stony Brook Center for Healthy Aging Award to NCR. We wish to acknowledge Dr. Oleksi Petrenko for his help in data analysis and advice with the investigation and manuscript, and Dr. Victoria Mingione for cell growth analyses. We also wish to thank Dr. Jinyu Li for assistance with data analysis, Jean Rooney in the Stony Brook University Division of Laboratory Animal Research for technical assistance in mouse surgeries, Yan Ji and technical support provided by the Research Histology Core Laboratory in the Department of Pathology, Dr. Richard Kew, Scientific Director of Stony Brook Medicine's Biobank in the Department of Pathology and Cancer Center, and Dr. Jonathan Preall, Head of Genomics Technology Development Cold Spring Harbor Laboratory (CSHL) for single-cell RNA-seq and bioinformatics analyses. We very much appreciate the generosity of Dr. David Tuveson (CSHL) and Dr. Ben Stanger (University of Pennsylvania) for providing KPC cells lines, and Dr. Marina Pasca di Magliano (University of Michigan) for providing iKRAS cell lines. Images for the schematic summary were derived from BioRender.

## Author contributions

**Stephen D'Amico**: Conceptualization; Data curation; Formal analysis; Investigation; Methodology; Writing—original draft. **Varvara Kirillov**: Formal analysis; Investigation; Methodology. **Jingxuan Liu**: Formal analysis; Investigation. **Zhijuan Qiu**: Formal analysis; Investigation. **Xinyuan Lei**: Formal analysis; Investigation. **Hong Qin**: Formal analysis; Investigation; Methodology. **Brian S Sheridan**: Formal analysis. **Nancy C Reich**: Conceptualization; Formal analysis; Supervision; Funding acquisition; Investigation; Writing—original draft; Project administration; Writing—review and editing.

Source data underlying figure panels in this paper may have individual authorship assigned. Where available, figure panel/source data authorship is listed in the following database record: biostudies:S-SCDT-10_1038-S44319-025-00563-w.

## Disclosure and competing interests statement

The authors declare no competing interests.

# Expanded View Figures

**Figure EV1. STAT3 and KRAS dependency.**

(**A**) Gene expression profiling of 168 human pancreatic adenocarcinomas in the TCGA PanCancer Atlas database (http://www.cbioportal.org) corresponding to gene signatures of KRAS dependency (KRAS-type) or reduced KRAS dependency (RSK-type)(Yuan et al, 2018). STAT3 mRNA expression z-score for each tumor sample is displayed on top. (**B**) Scatter plot showing correlation of two independently derived KRAS dependency gene signatures and their references for 150 human pancreatic adenocarcinomas in the TCGA database (http://www.cbioportal.org)(Singh and Settleman, 2009; Yuan et al, 2018). (**C**) Scatter plot showing correlation of STAT3 mRNA expression and RSK-type gene signature (Yuan et al, 2018) in human pancreatic ductal adenocarcinomas from the AUMC (Amsterdam University Medical Centers) database ($n = 80$). (**D**) Proteomic data from human pancreatic adenocarcinomas of the NCI Clinical Proteomic Tumor Analysis Consortium (CPTAC) comparing levels of STAT3 protein in TCGA tumors classified as KRAS-type or RSK-type ($n = 40$). Boxplots show center line as median, box limits as upper and lower quartiles, and whiskers the 1.5 interquartile range. Significance determined using two-tailed $t$ test. (**E**) Histo-score (H-score) of the cancer cells in 22 human pancreatic adenocarcinoma tumors from the Stony Brook Medicine Biobank following immunohistochemistry staining for phosphorylated-ERK (P-ERK). (**F**) Representative images of IHC staining of human pancreatic tumors scored in Fig. EV1E with antibodies to phospho-ERK (P-ERK) or STAT3. Tumor cancer cells (Ca) and fibroblast stromal cells (F) are indicated in a tumor. Scale bar 100 mm. (**G**) Representative sequences of major deletions created by CRISPR-mediated editing of mutant KRAS in the human PANC-1 cell line, determined following PCR amplification and cloning. (**H**) Growth in culture of PANC-1 parental cells expressing empty vector or knockout derivatives shown. Strip plots display doubling time for biological replicates ($n = 3$–8) with triplicate technical replicates. Significance was determined using two-tailed test at the 0.05 confidence interval. Plots show median center line and upper and lower quartile lines. (**I**) Representative sequences of the major indels modified by CRISPR-mediated gene editing of STAT3 and mutant KRAS in the KPC cell lines were determined following PCR amplification and cloning. (**J**) Growth in culture of KPC parental cells (PRTL) and derived STAT3 KO, KRAS KO, and DKO4 cells. Cell number during course of two weeks is shown ($n = 3$ biological replicates with triplicate technical replicates). (**K**) Relative tumor development during 60 days by parental KPC cells and derived STAT3 KO, KRAS KO and four DKO KPC cell lines following subcutaneous implantation in nude mice ($n = 8$–12 biological replicates). Data are presented as mean ± SD, two-tailed $t$ test. Source data are available online for this figure.

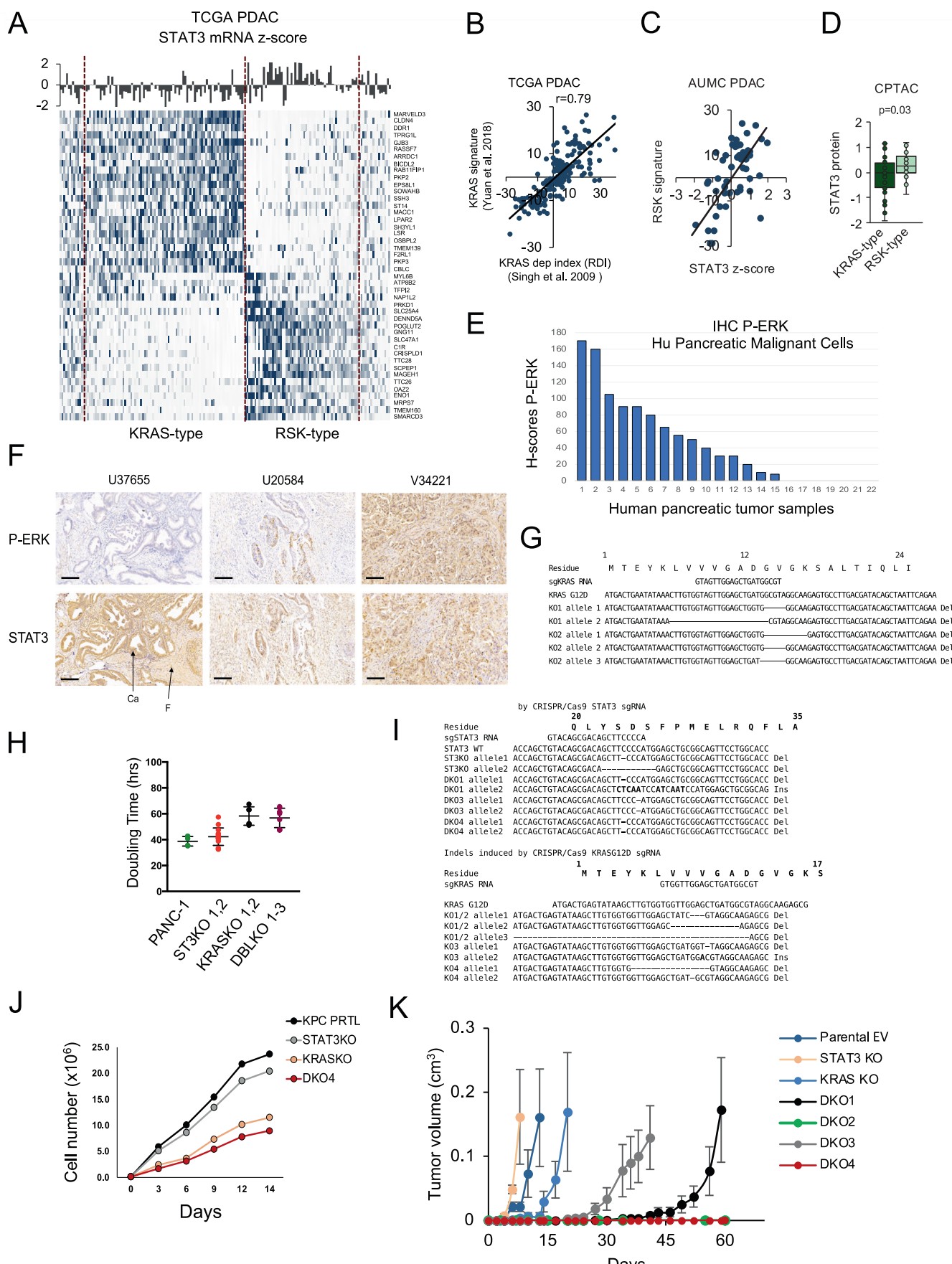

A

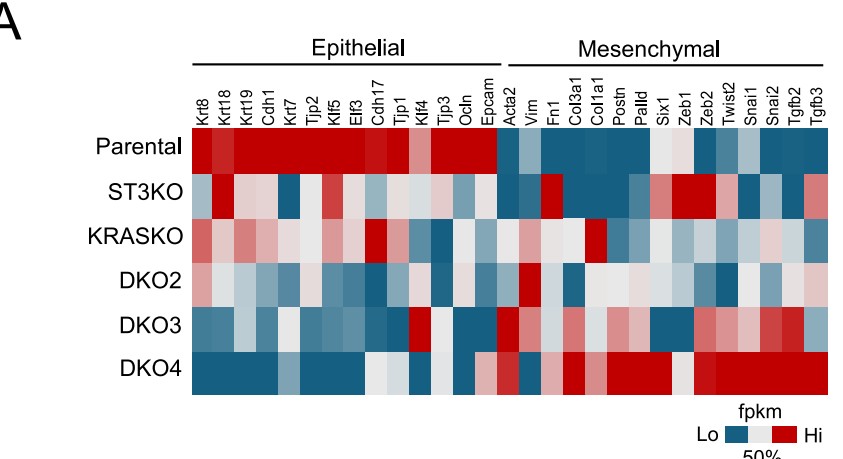

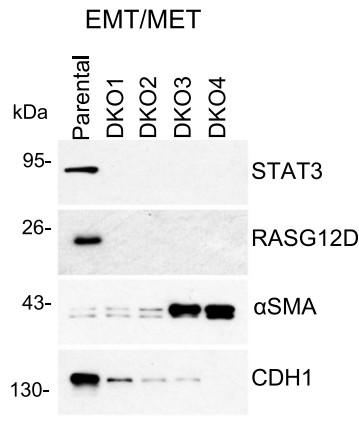

EMT/MET

B

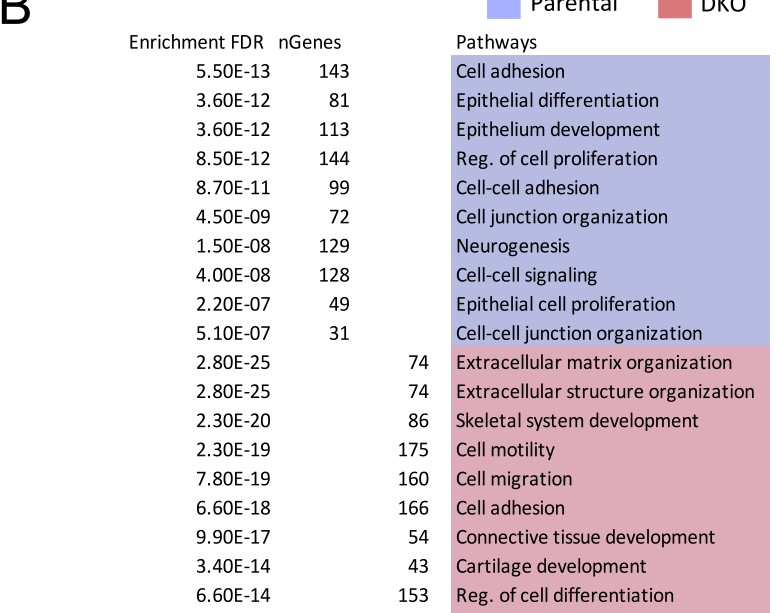

| Enrichment FDR | nGenes | | Pathways |
|---|---|---|---|
| 5.50E-13 | 143 | | Cell adhesion |
| 3.60E-12 | 81 | | Epithelial differentiation |
| 3.60E-12 | 113 | | Epithelium development |
| 8.50E-12 | 144 | | Reg. of cell proliferation |
| 8.70E-11 | 99 | | Cell-cell adhesion |
| 4.50E-09 | 72 | | Cell junction organization |
| 1.50E-08 | 129 | | Neurogenesis |
| 4.00E-08 | 128 | | Cell-cell signaling |
| 2.20E-07 | 49 | | Epithelial cell proliferation |
| 5.10E-07 | 31 | | Cell-cell junction organization |
| 2.80E-25 | | 74 | Extracellular matrix organization |
| 2.80E-25 | | 74 | Extracellular structure organization |
| 2.30E-20 | | 86 | Skeletal system development |
| 2.30E-19 | | 175 | Cell motility |
| 7.80E-19 | | 160 | Cell migration |
| 6.60E-18 | | 166 | Cell adhesion |
| 9.90E-17 | | 54 | Connective tissue development |
| 3.40E-14 | | 43 | Cartilage development |
| 6.60E-14 | | 153 | Reg. of cell differentiation |
| 2.10E-13 | | 155 | Neurogenesis |

Parental ☐ DKO ☐

C

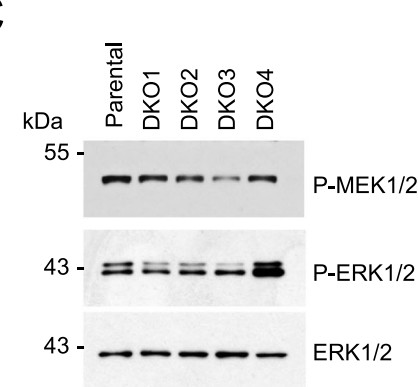

D

Transcription factors

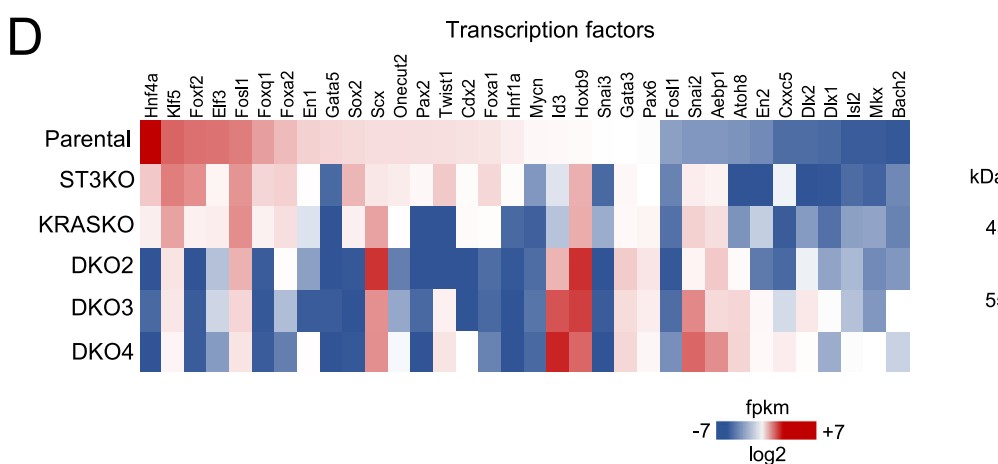

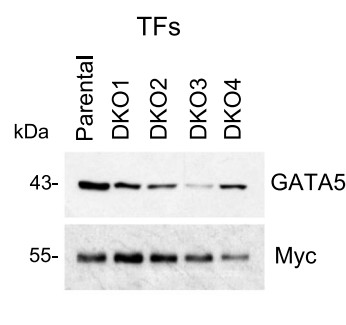

TFs

**Figure EV2.  Transcriptional reprogramming following loss of KRAS and STAT3.**

(A) Left) Heatmap of gene signatures for epithelial and mesenchymal identity derived from RNA-seq data of KPC parental, STAT3 KO cells, KRAS KO cells, and three of the DKO clones. Differences are compared for an individual gene across multiple cell types due to variance in gene representation. Highest (Hi) to lowest (Lo) counts (fpkm) and 50 percentile are shown for each gene. Right) Western blots of KPC parental cells and four DKO cell lines confirming RNA-seq data for expression of epithelial E-cadherin (CDH1) and mesenchymal smooth muscle actin (aSMA/ACTA2). (B) Gene ontology (GO) classification of top biological processes that are upregulated in KPC parental cells (blue) or DKO4 (red) cells. (C) Western blot of KPC parental cells and four DKO derived clones for phosphorylation activity of MEK and ERK1/2. Samples were run on the same gel as Fig. 2E with ERK1/2 control. (D) Left) Heatmaps of differential gene expression from RNA Seq analyses corresponding to a set of transcription factors in parental KPC cells, STAT3 KO cells, KRAS KO and three DKO derived clones. Right) Western blots of KPC parental cells and four DKO cell lines for GATA5, and MYC. Samples were run on the same gel as Fig. 2E. Source data are available online for this figure.

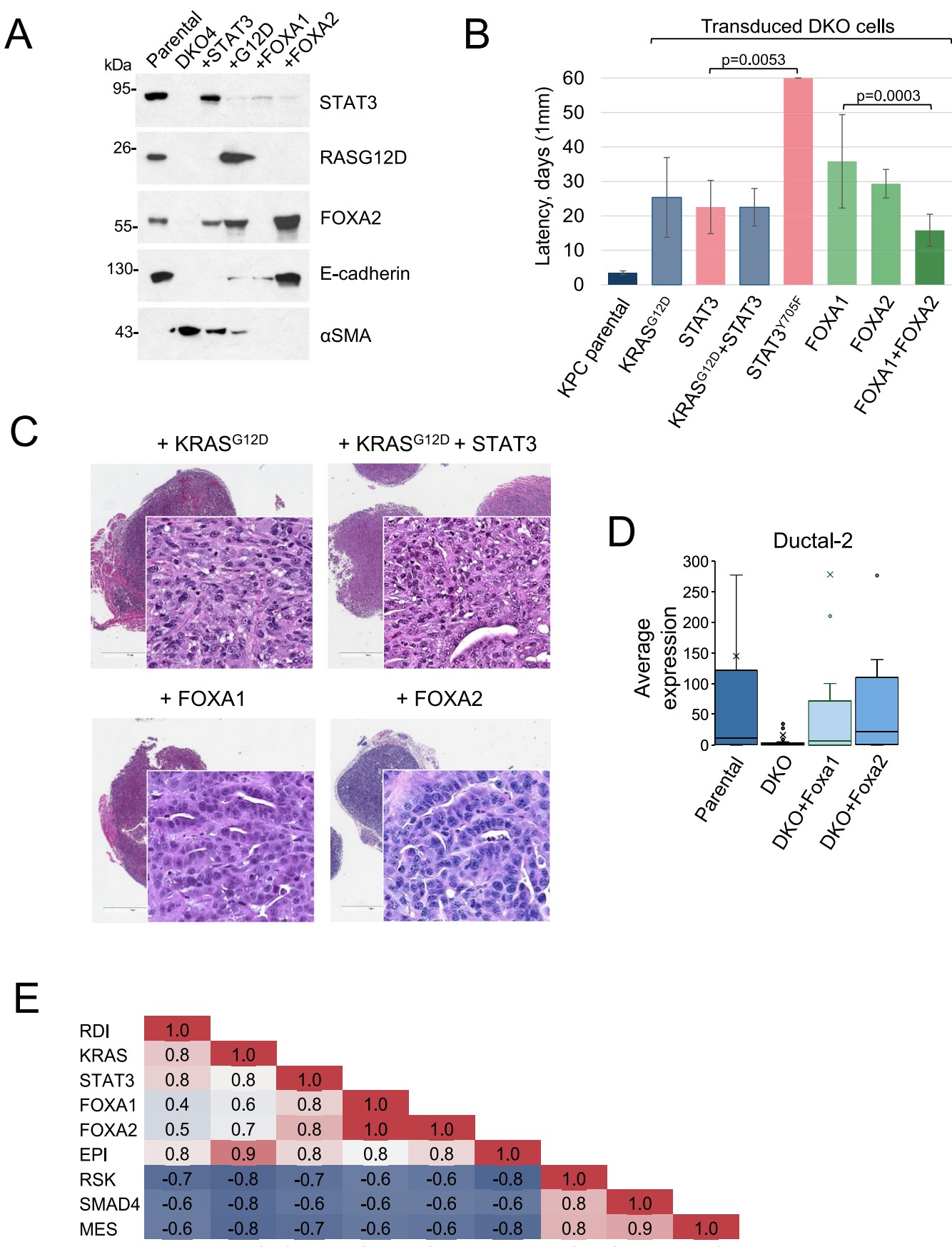

**Figure EV3. Transcriptional reprogramming following loss of KRAS and STAT3.**

(A) Representative Western blot of parental KPC cells, DKO cells, and DKO cells restored for expression of the designated gene by lentiviral transduction. (B) Restoration of subcutaneous tumorigenicity in nude mice of DKO4 cells quantified by tumor latency in days (~1 mm size tumor) following transduction with *Kras*$^{G12D}$, *Stat3*, *Stat3*$^{Y705F}$, *Foxa1, Foxa2* or the indicated combinations compared with parental KPC cells ($n = 8$–12). No tumors were formed by DKO4 cells. Significance was determined using two-tailed test at the 0.05 confidence interval. (C) Representative H&E histology images of tumors formed following transduction of DKO4 cells with genes noted. (D) Average expression of the top ductal-2 type PDAC genes determined by RNA-seq of KPC parental cells, DKO cells, or DKO cells transduced with *Foxa1* or *Foxa2* ($n = 2$–4)(Peng et al, 2019). Significance was determined using two-tailed test at the 0.05 confidence interval. (E) Pearson's coefficient analyses aligning gene expression signatures of reconstituted KPC cells as noted with PDAC TCGA tumor samples ($n = 168$, stage I/II tumors) according to RAS dependency (RDI), molecular subtype (i.e., KRAS-type vs RSK-type), and epithelial (EPI) or mesenchymal (MES) status. Source data are available online for this figure.

# A    Gene Ontology (GO) gene ratio

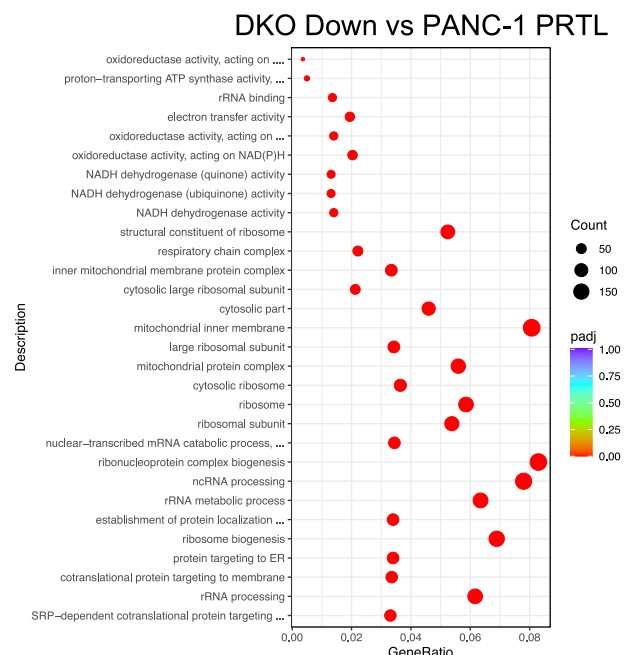

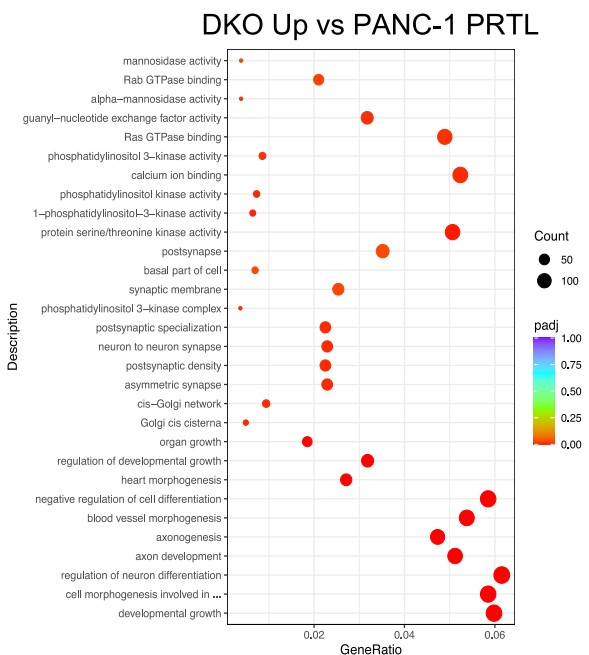

# B

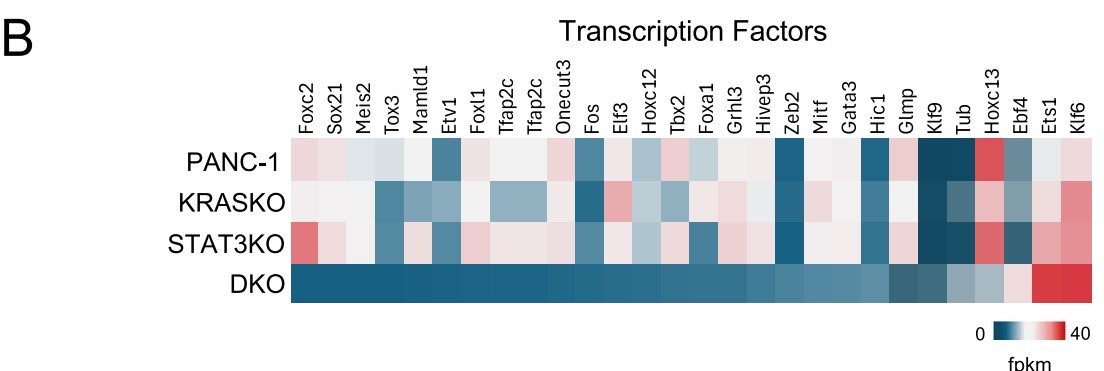

Transcription Factors

# C

Compare
Narrow peak fold-enrichment

| | KRAS KO | KPC Prtl | | |
|---|---|---|---|---|
| *Stat3* | 97 | 50 | -388 | ENSMUSG00000004040 |
| *Fos* | 95 | 34 | -323 | ENSMUSG00000021250 |
| *Twf1* | 82 | 23 | -285 | ENSMUSG00000022451 |
| *Rasa3* | 76 | 15 | 53503 | ENSMUSG00000031453 |
| *Jak3* | 75 | 35 | -1785 | ENSMUSG00000031805 |
| *Syt12* | 75 | 16 | -623 | ENSMUSG00000049303 |
| *Btd* | 66 | 25 | 21564 | ENSMUSG00000021900 |
| *Plec* | 64 | 24 | 7670 | ENSMUSG00000022565 |
| *Mir6979* | 55 | 24 | -25658 | ENSMUSG00000098706 |
| *Olfm3* | 42 | 18 | 124298 | ENSMUSG00000027965 |
| *Lsm4* | 41 | 24 | 265 | ENSMUSG00000031848 |
| *Cyp2c29* | 17 | 47 | -19830 | ENSMUSG00000003053 |

ChIP-seq anti-phospho Tyr705 STAT3

**Figure EV4. Transcriptional reprogramming following loss of KRAS and STAT3.**

(A) Gene ontology (GO) comparison of biological processes expressed in human PANC-1 DKO cells versus parental PANC-1 cells derived from RNASeq analyses. Comparisons shown as dot matrices. (B) Heatmaps of differential gene expression from RNA Seq analyses corresponding to a set of transcription factors in parental human PANC-1 EV cells or derived KRAS KO, STAT3 KO, or DKO cells. (C) Differential enrichment of genes in KRASKO and KPC parental cells identified by ChIP with antibodies to tyrosine 705 phosphorylated STAT3. Comparisons are shown as fold enrichment in narrow peaks for genes relative to control antibody or DKO cells (count per 1 M reads IP/input). Gene identity and location of the transcriptional start sites are indicated.

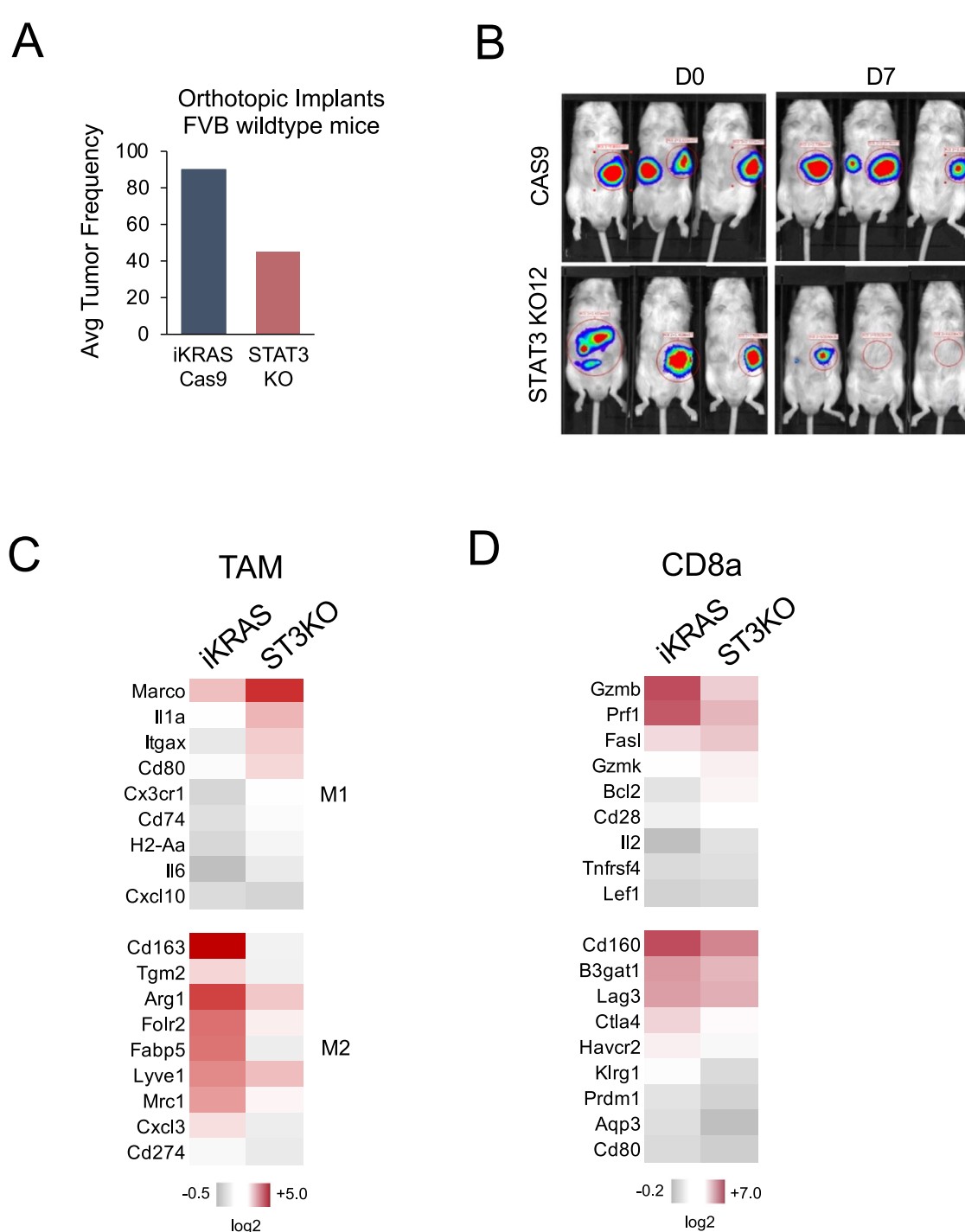

**Figure EV5.  STAT3 depletion in an inducible mutant KRAS PDAC model.**

(A) Average frequency of tumor formation in FVB wild-type mice treated with doxycycline of parental iKRAS cells expressing empty vector (EV) or three independent CRISPR-edited STAT3 KO iKRAS clones. (B) Representative in vivo bioluminescence imaging of tumors formed by orthotopic implants of iKRAS parental cells expressing empty vector (Cas9) or derived STAT3 KO12 cells that were transduced to express luciferase. Mice were administered doxycycline and tumors were evaluated on day 0 (D0) or seven days after withdrawal of doxycycline (D7). (C) Comparative heatmaps of scRNA-seq individual datasets for a subset of genes expressed in the tumor associated macrophages (TAMs) in tumors formed by iKRAS control cells or STAT3KO derived cells in FVB mice treated with doxycycline based on captured viable cell populations and proportions. (D) Comparative heatmaps of scRNA-seq individual datasets for a subset of genes expressed in CD8a+ T cells (CD8a expression log2 fold greater than CD4 expression) in tumors formed by iKRAS control cells or STAT3KO derived cells in FVB mice treated with doxycycline based on captured viable cell populations and proportions. Source data are available online for this figure.

