## [Peer Review File · EMBO Reports]

STAT3 Sustains Tumorigenicity Following Mutant KRAS Ablation

Stephen D'Amico, Varvara Kirillov, Jingxuan Liu, Zhijuan Qiu, Xinyuan Lei, Hong Qin, Brian Sheridan, and Nancy Reich

Corresponding author(s): Nancy Reich (nancy.reich@stonybrook.edu)

Review Timeline:

Submission Date:	14th Mar 25
Editorial Decision:	29th Apr 25
Revision Received:	16th Jun 25
Editorial Decision:	10th Jul 25
Revision Received:	4th Aug 25
Accepted:	14th Aug 25

Editor: Achim Breiling

Transaction Report:

Dear Dr. Reich,

Thank you for the transfer of your manuscript to EMBO reports. I have now received the reports from the three referees that were asked to evaluate your study, which can be found at the end of this email.

As you will see, the referees think that these findings are of interest. However, they have several comments, concerns, and suggestions, indicating that a major revision of the manuscript is necessary to allow publication of the study in EMBO reports. As the reports are below, and all the referee concerns need to be addressed, I will not detail them here.

Given the constructive referee comments, I would like to invite you to revise your manuscript with the understanding that the concerns of the referees must be addressed in the revised manuscript and/or in a detailed point-by-point response. Acceptance of your manuscript will depend on a positive outcome of a second round of review. It is EMBO reports policy to allow a single round of revision only and acceptance of the manuscript will therefore depend on the completeness of your responses included in the next, final version of the manuscript.

- 1) a .docx formatted version of the final manuscript text (including legends for main figures, EV figures and tables), but without the figures included. Figure legends should be compiled at the end of the manuscript text.
- 2) individual production quality figure files as .eps, .tif, .jpg (one file per figure), of main figures and EV figures. Please upload these as separate, individual files upon re-submission.

- 4) a complete author checklist, which you can download from our author guidelines (<https://www.embopress.org/page/journal/14693178/authorguide>). Please insert page numbers in the checklist to indicate where the requested information can be found in the manuscript. The completed author checklist will also be part of the RPF.

- 5) that primary datasets produced in this study (e.g. RNA-seq, ChIP-seq, structural and array data) are deposited in an

appropriate public database. If no primary datasets have been deposited, please also state this in a dedicated section (e.g. 'No primary datasets have been generated and deposited'), see below.

The accession numbers and database should be listed in a formal "Data Availability" section that follows the model below. This is now mandatory (like the COI statement). Please note that the Data Availability Section is restricted to new primary data that are part of this study. This section is mandatory. As indicated above, if no primary datasets have been deposited, please state this in this section

Data availability

6) We now request the publication of original source data with the aim of making primary data more accessible and transparent to the reader. You will receive a separate email with instructions for providing source data with your revised manuscript, including information how to upload and organize the files.

8) Regarding data quantification and statistics, please make sure that the number "n" for how many independent experiments were performed, their nature (biological versus technical replicates), the bars and error bars (e.g. SEM, SD) and the test used to calculate p-values is indicated in the respective figure legends (also for EV and Appendix figures). Please also check that all the p-values are explained in the legend, and that these fit to those shown in the figure. Please provide statistical testing where applicable. Please avoid the phrase 'independent experiment', but clearly state if these were biological or technical replicates. Please also indicate (e.g. with n.s.) if testing was performed, but the differences are not significant. In case n=2, please show the data as separate datapoints without error bars and statistics. See also: <http://www.embopress.org/page/journal/14693178/authorguide#statisticalanalysis>

9) Please add scale bars of similar style and thickness to microscopic images, using clearly visible black or white bars (depending on the background). Please place these in the lower right corner of the images themselves. Please do not write on or near the bars in the image but define the size in the respective figure legend.

10) Please also note our reference format:

12) We now use CRedit to specify the contributions of each author in the journal submission system. CRedit replaces the author contribution section. Please use the free text box to provide more detailed descriptions and do NOT provide your final manuscript text file with an author contributions section. See also our guide to authors: <https://www.embopress.org/page/journal/14693178/authorguide#authorshipguidelines>

13) All Materials and Methods need to be described in the main text using our 'Structured Methods' format, which is required for

all research articles. According to this format, the Methods section should include a Reagents and Tools Table (listing key reagents, experimental models, software, and relevant equipment and including their sources and relevant identifiers), uploaded as separate file, and a Methods section in which we encourage the authors to describe their methods using a step-by-step protocol format with bullet points, to facilitate the adoption of the methodologies across labs. More information on how to adhere to this format as well as downloadable templates (.doc) for the Reagents and Tools Table can be found in our author guidelines (section 'Structured Methods'):

14) Please order the sections like this, using (only) these names:

Title page - Abstract - Keywords - Introduction - Results - Discussion - Methods - Data availability section - Acknowledgements - Disclosure and Competing Interests Statement - References - Figure legends - Expanded View Figure legends

Please fuse the Abstract and the Significance section (with not more than 175 words).

15) Please make sure that all the funding information is also entered into the online submission system and that it is complete and similar to the one in the acknowledgement section of the manuscript text file.

Please note that corresponding authors are required to supply an ORCID ID upon submission of a revised manuscript and an institutional e-mail address. Please make sure that the corresponding author provides an ORCID in the submission system. Please find instructions on how to link the ORCID ID to the account in our manuscript tracking system in our author guidelines: <http://www.embopress.org/page/journal/14693178/authorguide#authorshipguidelines>

I look forward to seeing a revised form of your manuscript when it is ready.

Please use this link to submit your revision: <https://embor.msubmit.net/cgi-bin/main.plex>

Yours sincerely,

Referee #1:

Pancreatic ductal adenocarcinoma (PDAC) is a highly aggressive malignancy characterized by rapid progression and resistance to treatment. Although KRAS mutations drive over 90% of PDAC cases, direct KRAS inhibition has achieved limited clinical success, partly due to tumor heterogeneity and adaptive resistance mechanisms. Emerging evidence suggests that even potent KRAS inhibition may not cure PDAC. STAT3, a signal transducer and activator of transcription, has been implicated as a key regulator that sustains malignancy following KRAS inactivation. While the role of STAT3 in early tumorigenesis remains context-dependent, this study investigates its function in maintaining pancreatic cancer tumorigenicity after KRAS loss.

Major comments:

1. The rescue experiments using FOXA1/2 are presented. However, it remains unclear whether FOXA1/2 are directly transcriptionally regulated by STAT3 or if this regulation is indirect. ChIP-seq or promoter analysis data (even from prior literature) would strengthen this link.
2. The manuscript does not address how STAT3 becomes activated following KRAS ablation. A brief discussion or preliminary data on potential upstream activators (e.g., cytokines such as IL-6) would enhance the mechanistic understanding.
3. To increase the clinical relevance of the findings, the authors should evaluate the effect of pharmacological STAT3 inhibition in the KRAS KO models. Even preliminary in vivo or in vitro results would significantly strengthen the translational potential of the study.
4. The authors should provide evidence that STAT3 is phosphorylated (activated) in the KRAS-mutant tumors. Western blotting

or immunohistochemistry for phosphorylated STAT3 (p-STAT3 Y705) would validate its functional activation status in these models..

5. The authors suggest that DKO cells exhibit features associated with epithelial plasticity based on changes in gene expression (Figure 2G). However, the genes presented are not canonical markers of the epithelial-mesenchymal transition (EMT) program, such as CDH1, VIM, SNAI1, or ZEB1. To more convincingly demonstrate that the DKO cells acquire EMT characteristics, additional analyses are needed. Specifically, the authors should evaluate whether the DKO cells display functional properties associated with EMT, such as increased migration or invasion capabilities, and whether they undergo morphological changes consistent with a mesenchymal phenotype. Furthermore, gene set enrichment analysis (GSEA) of EMT hallmark pathways using their RNA-seq data could provide a more systematic and quantitative assessment. Strengthening this aspect would clarify the nature of the cellular reprogramming induced by combined KRAS and STAT3 loss.

Minor comments:

1. Figure 2B appears misaligned or improperly labeled. Please correct.
2. The UMAP plot in Figure 4D is colored based on cell types, but the method of annotation is not explained. The authors should specify whether this was based on canonical marker genes or automated tools (e.g., SingleR, Seurat annotations), and if so, which markers or method was used.
3. Given that tumors from 2-3 mice were pooled for single-cell RNA-seq, a supplementary UMAP plot colored by donor ID would be helpful to assess and rule out batch effects or donor bias in clustering.
4. The authors should be cautious when interpreting cellular proportions derived from scRNA-seq. Factors such as dissociation artifacts, capture bias, and variable cell viability may skew results, especially under-representing epithelial cells relative to immune cells. The data should be described as estimates of "captured viable cell proportions" rather than true tissue proportions (e.g., modify the statement around line 283).

Referee #2:

In this manuscript, the authors explore an interesting topic and have likely uncovered important findings regarding the requirement for STAT3 in KRAS-depleted conditions. However, enthusiasm for this manuscript was reduced by the incomplete nature in which key experiments were presented.

Major concerns:

1. The sentences in the Abstract stating, "STAT3 is not an oncogene, but its inactivation is needed to ensure that KRAS ablated tumor cells lose their malignant identity. Mechanistically, the combined loss..." are discussion, rather than description of results as would be expected in an abstract.
2. The manuscript would be strengthened if authors would provide more context in the introduction, eg in paragraph starting on line 65. An enormous knowledge base exists about both KRAS and STAT3, but this context does not come through for the reader as presented.
3. Starting on line 74, the authors summarize their own findings. This should be within the abstract and discussion, not in the introduction.
4. Can't read labels in multiple figures including Fig 1A, 4E, 4F.
5. Experimental design and set-up was problematically vague and/or incompletely described in many places. Some examples are indicated here:
 - a. Line 113: how was gene expression analyzed?
 - b. Line 119: what samples? Where is the data proving this point?
 - c. Line 200: what is the definition of "latency" here?
 - d. Authors should show western blot validation of overexpression and latency graph in main figure instead of table Fig 2F.
 - e. Fig 2G: more clear experimental design description would facilitate interpretation
 - f. Line 240: where is quantitation proving ">100 fold"?
 - g. Starting line 241: experimental design of mouse experiment is confusing
 - h. Line 245: control mice should have been treated with vehicle instead of "untreated"
 - i. Line 245: what is definition of "KRAS inhibitory cocktail?" how was it administered?
 - j. Fig 3C needs to be quantitated.
 - k. Fig 3 overall is confusing and significantly underdeveloped.
 - l. Top paragraph on page 11 is confusing.
 - m. Line 283: what is definition of "malignant" tumor cells and how was this proven here?
 - n. scRNA-seq analysis is overly simplistic.

Referee #3:

D'Amico and colleagues explore the molecular underpinnings of pancreatic ductal adenocarcinoma (PDAC). Their experiments are largely based on the deletion of endogenous KRAS and/or STAT3 in two different cell lines. One is PANC1, human cells that express activated KRAS (G12D) but are considered nonetheless a model of KRAS-independent tumor cells. The other is the murine KPC cell line, which expresses the same activating KRAS mutant and a p53 gain-of-function mutant (R172H). Both cell lines are models for pancreatic ductal adenocarcinoma. The authors characterize these cells in a number of ways including their implantation in nude and wild type (B16) mice (subcutaneous and into pancreas), which convincingly shows that cells devoid of both KRAS and STAT3 expression are devoid too of tumor-inducing activity. Based on these interesting observations, they do a number of additional meaningful experiments (transcription profiling; STAT3/KRAS reconstitution followed by transplantation into mice; pharmacological KRAS inhibition; single cell RNASeq experiments using an inducible KRAS mouse strain) to attempt to gain insights into the molecular mechanisms by which STAT3 contributes to tumor growth in the absence and presence of mutant KRAS.

Overall, this is well-executed work that convincingly confirms current understanding that STAT3 contributes to tumor growth and behaviors in PDAC. What is missing, however, is further clarification of how STAT3 contributes. The consequences caused by the approach chosen here, namely complete removal of STAT3, can be interpreted in several ways. (i) STAT3 is required as a protein, possibly in a scaffolding role; (ii) alternatively, its activation by Tyr-phosphorylation is necessary and hence transcription factor function; or (iii) what is observed are compensatory effects caused by another protein that fills the void created by the removal of STAT3. An answer to this question would go beyond the current understanding, and an attempt should be made to provide further insights. This seems pertinent in light of the reported results, such as the substantial transcriptional deregulation caused by STAT3 deletion shown in Fig. 2. In Fig 3 it is shown that KRAS inhibition leads to PDAC tumor regression. It remains unclear if the massive increase in STAT3 activation that accompanies this process contributes to tumorigenesis at all, and if yes, in pro- or anti-tumorigenic capacity. Also, in Fig 2, the re-expression of STAT3 partially restored tumorigenicity in nude mice, but if that can be achieved with a non-activatable STAT3 mutant has not been assessed.

Answers to these questions are not only important to understand STAT3 cell biology, but are also relevant from a clinical point of view, namely if it suffices to suppress STAT3 activation or if the STAT3 protein needs to be removed to inhibit tumor growth.

EMBOR-2025-61534**Response to Review**

We would like to thank the Referees for their time and valuable insight. We have carefully considered each of their comments and revised and improved the manuscript.

Changes to the text and responses to referee's comments are highlighted in blue.

Referee #1:

Major comments:

1. The rescue experiments using FOXA1/2 are presented. However, it remains unclear whether FOXA1/2 are directly transcriptionally regulated by STAT3 or if this regulation is indirect. ChIP-seq or promoter analysis data (even from prior literature) would strengthen this link.

We did investigate the nature of STAT3 regulation of FOXA1/2 and have now added information to the manuscript (page 10). STAT3-ChIP-seq datasets encompass thousands of genes and vary based on different cell types that are influenced by chromatin accessibility and specific transcription factor expression. Integration of these results with the Harmonizome database identifies *FoxA1* as a specific target of STAT3, and we have added this information to the text. (https://maayanlab.cloud/Harmonizome/gene_set/STAT3/ENCODE+Transcription+Factor+Targets) [I. Diamant et al. 2025].

We also performed a stringent chromatin immunoprecipitation set of experiments with control antibodies or antibodies to phosphotyrosine 705 STAT3 in the KPC parental cells, isogenic KRAS knockout cells, and isogenic KRAS/STAT3 double knockout cells as a control. The results identified previous STAT3 gene targets as well as new targets, but did not identify *Foxa1* or *Foxa2*, possibly signifying an indirect regulation. Importantly, some of the target genes have a significantly increased fold enrichment in the KRAS knockout cells compared with KPC parental cells. The increased representation of these genes in the KRAS knockout cells may reflect the ability of STAT3 to support KRAS independence. We have now included this information in the manuscript and added Supplementary Fig EV2L.

2. The manuscript does not address how STAT3 becomes activated following KRAS ablation. A brief discussion or preliminary data on potential upstream activators (e.g., cytokines such as IL-6) would enhance the mechanistic understanding.

We agree with the Reviewer and have now added information to the Discussion (pg15).

In our previous publication we evaluated scRNASeq of tumors generated by KPC parental cells and isogenic KRAS knockout cells (Ischenko et al 2021). Expression of the IL-6 cytokine by cancer associated fibroblasts (CAFs) increased in the KRAS knockout tumors relative to the control KPC tumors and could be responsible for increased STAT3 tyrosine phosphorylation. In our current manuscript, RNASeq showed more than a 5-fold increase in IL-11 expression by the KRAS knockout PDAC cells. In addition, the scRNASeq data of iKRAS tumor cells showed an increase in IL-6 and IL-11 receptors following KRAS depletion, indicating their increased responsiveness to STAT3 activating cytokines.

A previous publication by Miyazaki et al. 2024 demonstrated induction of STAT3 tyrosine phosphorylation following inhibition of KRAS and MEK with MRTX133 and trametinib in human pancreatic cancer cells. This increase in STAT3 tyrosine phosphorylation was ablated with the JAK inhibitor fedratinib.

Therefore, genetic or drug depletion of mutant KRAS in PDAC appears to increase tyrosine phosphorylation of STAT3 by means of JAK activation by cytokines of the IL-6 family.

3. To increase the clinical relevance of the findings, the authors should evaluate the effect of pharmacological STAT3 inhibition in the KRAS KO models. Even preliminary in vivo or in vitro results would significantly strengthen the translational potential of the study.

Clinical targeting of STAT3 remains to be actualized, although some small molecules, oligonucleotides, peptides, and PROTACs are in early phase 1 trials [W. Wang et al. 2024]. Targeting upstream JAKs can decrease STAT3 tyrosine phosphorylation, however JAK inhibitors suppress the development and function of immune cells such as T and B lymphocytes. For these reasons we used direct genetic depletion of STAT3 to demonstrate its contribution to tumorigenicity following KRAS drug induced or genetic ablation.

We tested the effect of JAK inhibitors such as tofacitinib alone or in conjunction with KRAS and MEK inhibitors, but they proved ineffective at reducing tumor mass.

The development and screening of clinical pharmacological inhibitors of STAT3 is outside the purview of this study.

4. The authors should provide evidence that STAT3 is phosphorylated (activated) in the KRAS-mutant tumors. Western blotting or immunohistochemistry for phosphorylated STAT3 (p-STAT3 Y705) would validate its functional activation status in these models.

Immunohistochemistry of KPC mutant tumors with anti-phosphotyrosine 705 STAT3 antibodies is presented in Figure 3. We have now replaced the images in this figure with similar tumor sections, but the result is the same. There is a low level of STAT3 tyrosine phosphorylation in the pancreatic ductal carcinoma, and this increases in response to KRAS pathway inhibition.

Further support of STAT3 activity in human pancreatic cancers is provided by phosphoproteomic analyses by the Clinical Proteomic Tumor Analysis Consortium (CPTAC) that indicate low to moderate levels of STAT3 Y705 phosphorylation [Cao et al., Cell, 2021 Sep 16;184(19):5031–5052.e26. doi: [10.1016/j.cell.2021.08.023](https://doi.org/10.1016/j.cell.2021.08.023)

5. The authors suggest that DKO cells exhibit features associated with epithelial plasticity based on changes in gene expression (Figure 2G). However, the genes presented are not canonical markers of the epithelial-mesenchymal transition (EMT) program, such as CDH1, VIM, SNAI1, or ZEB1. To more convincingly demonstrate that the DKO cells acquire EMT characteristics, additional analyses are needed. Specifically, the authors should evaluate whether the DKO cells display functional properties associated with EMT, such as increased migration or invasion capabilities, and whether they undergo morphological changes consistent with a mesenchymal

phenotype. Furthermore, gene set enrichment analysis (GSEA) of EMT hallmark pathways using their RNA-seq data could provide a more systematic and quantitative assessment. Strengthening this aspect would clarify the nature of the cellular reprogramming induced by combined KRAS and STAT3 loss.

Thank you for the opportunity to expand on the contribution of KRAS and STAT3 to the maintenance of epithelial identity in PDAC.

Previously we published that loss of STAT3 or loss of mutant KRAS in PDAC cells leads to an increase in EMT gene signatures and an increase in mesenchymal tumor morphology (D'Amico et al 2018; Ischenko et al 2021; D'Amico et al 2023).

In addition, publications by others with tumors or tumor cell lines also revealed a loss of epithelial identity and a quasi-mesenchymal gene signature in response to KRAS depletion (Muzumder et alref,; Singh et al 2009, Yuan et al 2018).

In this study we analyzed gene expression by RNA-seq in multiple independent experiments and find loss of STAT3 and KRAS reduces epithelial identity and promotes a mesenchymal-like identity.

In Figure 2B we compared RNA-seq results of parental and knockout cells expressing twenty of the top EMT genes listed in GSEA/Molecular Signatures Database and showed a box plot comparison.

The specific EMT gene fpkm expression used for this graph:

EMT	Parental	STAT3KO	KRASKO	DKO
Acta2	0.087473	0.0475	189.1169	897.5052
Sparc	3.206109	2.004606	304.3025	685.7314
Mgp	0.347357	0.141468	11.49189	493.2661
Tnc	0.665019	9.286022	215.2584	486.7151
Gas1	2.50793	111.2974	32.93451	318.074
Col6a2	0.623751	0.103336	65.50568	299.2528
MyI9	0.046037	0	34.16058	249.8659
Crlf1	7.233549	2.032617	114.7548	197.6248
Postn	0.011919	0	91.96966	138.4142
Col5a1	6.363053	5.837124	70.76494	137.6638
Matn2	3.542327	4.339877	29.50422	83.83306
Dab2	3.830988	4.535602	31.72096	81.1468
Fstl1	1.363692	1.088455	30.31596	70.49609
Col1a1	0.411586	0.156927	81.06632	60.73762
Fbln5	0.356356	0.02177	16.91946	60.52312
Col11a1	1.388501	2.195775	33.90576	59.35405
Thbs1	0.911083	1.192011	27.15945	56.91365
Mest	1.951947	1.209017	25.78626	47.92673
Col16a1	0.418951	1.056257	15.44766	45.56926
Loxl2	6.359365	4.630931	33.01077	45.32451

In Fig EV2A we show a heatmap of RNA-Seq fpkm results for mesenchymal and epithelial gene expression for several DKO lines in comparison to parental KPC cells, STAT3 KO cells, and KRAS KO cells. This includes fpkm data for specific epithelial and mesenchymal genes below.

	Parental	STAT3KO	KRASKO	DKO
Epcam	2.0944748	0.26955259	0.14431424	0.65077074
Krt7	168.821303	0.89888003	14.1560698	2.2699087
Krt8	2021.07748	680.744746	1579.39698	0.88259656
Krt19	887.943998	395.103992	581.730286	3.93105171
Cdh1	188.16268	47.5983342	71.1271421	0.15982812
Elf3	55.4018039	8.27809006	10.8606184	1.73207526
Klf4	19.7306422	13.2804124	5.29013822	0.74508846
Acta2	0.75919669	0.04750031	56.9343816	1299.29649
Col1a1	0.29185928	0.15692723	29.5535818	15.3502881
Tgfb3	20.8892789	56.6236897	24.8689845	79.3378768
Zeb2	0.73593343	7.03668562	4.79507599	6.9440302
Twist2	1.4351313	3.95349267	1.94879338	6.41440883
Snai1	0.20019188	0	0.23350749	14.6052065
Snai2	1.62441925	10.7465755	20.9147543	59.7875996

In addition, in Fig EV 2A we show a Western blot for CDH1 and ACTA2/SMA.

In Fig EV 2B we display Gene Ontology (GO) classification of biological processes based on the RNA-Seq results comparing parental KPC cells with DKO4. The KPC parental cells expressing mutant KRAS are clearly more epithelial in nature.

In Figure 1J we have now enlarged the H&E histology inset so that the epithelial morphology of KPC parental tumors in comparison to the mesenchymal morphology of the STAT3 and KRAS KO tumors is more apparent.

Minor comments:

1. Figure 2B appears misaligned or improperly labeled. Please correct.

Thank you, we have realigned this figure.

2. The UMAP plot in Figure 4D is colored based on cell types, but the method of annotation is not explained. The authors should specify whether this was based on canonical marker genes or automated tools (e.g., SingleR, Seurat annotations), and if so, which markers or method was used.

The Seurat annotations provided by Cold Spring Harbor Laboratory Single-Cell Biology Shared Resource were used first to identify cell types. Then we used the Loupe program and canonical gene sets to include or exclude cell type and derived a simple color depiction.

Some of the genes used included *ONECUT2* (tumor cell), *TRAC* (T cell), *CD79* (B cell), *FAP* (fibroblast), *ADGRE1* (myeloid/macrophage), *CSF3R* (granulocyte/neutrophil), and *CD93* (endothelial cells).

We have now added this information to the Figure 4D legend.

3. Given that tumors from 2-3 mice were pooled for single-cell RNA-seq, a supplementary UMAP plot colored by donor ID would be helpful to assess and rule out batch effects or donor bias in clustering.

We pooled tumors from 2-3 individual mice for each sample analyzed by scRNA-Seq in order to decrease technical or animal variability. Since the samples were pooled tumors, individual data does not exist.

4. The authors should be cautious when interpreting cellular proportions derived from scRNA-seq. Factors such as dissociation artifacts, capture bias, and variable cell viability may skew results, especially under-representing epithelial cells relative to immune cells. The data should be described as estimates of "captured viable cell proportions" rather than true tissue proportions (e.g., modify the statement around line 283).

Thank you, we have now modified our statement describing the data in Figure 4 (pg 11).

Referee #2:

Major concerns:

1. The sentences in the Abstract stating, "STAT3 is not an oncogene, but its inactivation is needed to ensure that KRAS ablated tumor cells lose their malignant identity. Mechanistically, the combined loss..." are discussion, rather than description of results as would be expected in an abstract.

Thank you, we have now removed this sentence from the Abstract and have added the concept to the Discussion.

2. The manuscript would be strengthened if authors would provide more context in the introduction, eg in paragraph starting on line 65. An enormous knowledge base exists about both KRAS and STAT3, but this context does not come through for the reader as presented.

Thank you, we have rewritten the Introduction to include additional STAT3 background and rationale for the study.

3. Starting on line 74, the authors summarize their own findings. This should be within the abstract and discussion, not in the introduction.

We have revised the Introduction and removed much of the text describing the study.

4. Can't read labels in multiple figures including Fig 1A, 4E, 4F.

We have improved clarity of these figures and increased font size accordingly.

5. Experimental design and set-up was problematically vague and/or incompletely described in many places. Some examples are indicated here:

a. Line 113: how was gene expression analyzed?

We have now improved the description of our analyses in Materials & Methods:

“Human tumor samples were classified as KRAS dependent/KRAS-type, or KRAS independent/RSK-type by calculating the sum of individual mRNA expression values (z-scores) of genes that were previously characterized as KRAS dependent or independent and these are listed in the figures (Singh *et al.*, 2009; Yuan *et al.*, 2018). A murine KRAS dependency signature was used to calculate RAS dependency scores and was previously described (D’Amico *et al.*, 2024; Ischenko *et al.*, 2021). STAT3 signature scores were computed from a set of regulated genes that are listed in the figure (Dauer *et al.*, 2005).” (pg. 21)

We have now added information to the Results section.

“Tumors were classified by the sum of mRNA expression values (z-scores) of individual genes that comprise KRAS-dependent or KRAS-independent gene signatures.” (pg. 5)

b. Line 119: what samples? Where is the data proving this point?

We have improved the Materials & Methods section identifying the human PDAC samples:

“Human pancreatic adenocarcinoma tumor samples were obtained as paraffin-embedded tissue specimens from the Stony Brook Medicine BioBank [<https://renaissance.stonybrookmedicine.edu/pathology/biobank>].” (pg. 20)

The graphed data is presented in Supplementary Fig. 1E, withv examples of immunohistochemistry of tumors.

c. Line 200: what is the definition of "latency" here?

Tumor latency has been defined in the Materials & Methods as the period between implantation of tumorigenic cells and the appearance of tumors ~1mm for subcutaneous tumors and ~350mm³ for orthotopic tumors. (pgs. 21, 22)

d. Authors should show western blot validation of overexpression and latency graph in main figure instead of table Fig 2F.

The Table is presented in Fig. 2F because it provides more information regarding latency, frequency, and numbers of mice.

We have complemented the data with a Western blot of transduced DKO cells and overexpression of STAT3, KRASG12D and FOXA2 in Fig. EV 2E. We also show results as a latency graph of the parental, DKO, and transduced DKO cells in Fig. EV 2F.

e. Fig 2G: more clear experimental design description would facilitate interpretation

We have now included the sentences "Gene expression in DKO cells transduced with *Foxa1* or *Foxa2* was evaluated by RNA-Seq and compared with parental KPC or DKO cells. A subset of gene expression differences is presented in heatmaps (Fig. 2G)." (pg. 9)

f. Line 240: where is quantitation proving ">100 fold"?

I sincerely regret that this was an error in the manuscript text. It should have noted a 10-fold increase, and this has been corrected. (pg. 11)

We have also included an explanatory sentence in Materials & Methods: "Comparative immunohistochemistry staining for phosphorylated tyrosine 705 STAT3 (CST9145) and CD8a (CST98941) was performed with KPC tumors by visual scoring signal above background of four random microscopic fields." (pg. 20)

g. Starting line 241: experimental design of mouse experiment is confusing

We have improved the descriptive text and added a timeline image to Figure 3B for clarity. (pgs. 10, 11)

h. Line 245: control mice should have been treated with vehicle instead of "untreated"

The control mice were treated with rat IgG2a control antibodies (BioXCell) in vehicle and we have now corrected the text and the Figure 3A.

i. Line 245: what is definition of "KRAS inhibitory cocktail?" how was it administered?

We have improved the description of the drugs and antibodies administered by intraperitoneal injection (pgs. 10, 11)

j. Fig 3C needs to be quantitated.

Figure 3C is only a histological example of the data in Figure 3B.

k. Fig 3 overall is confusing and significantly underdeveloped.

We have now improved the description of the experiments in Figure 3. The results show that KPC tumor cells lacking STAT3 regress more readily in response to a KRAS inhibitor regiment for one week than KPC cells expressing STAT3. It provides proof of principle that STAT3

function supports resistance to a KRAS inhibitor regiment. Clinical targeting of STAT3 remains to be actualized and for these reasons we used direct genetic depletion of STAT3. (pgs. 10, 11)

l. Top paragraph on page 11 is confusing.

We have improved the description of these analyses.

m. Line 283: what is definition of "malignant" tumor cells and how was this proven here?

We used the term malignant to specify the pancreatic tumors cells within the tumor distinct from fibroblasts, immune cells, etc.. We did not intend to imply metastatic, and have now removed the adjective from the text.

n. scRNA-seq analysis is overly simplistic.

We provided a broad picture of the iKRAS scRNA-Seq data and not granular details of the many subtypes of tumor cells, myeloid cells, fibroblasts, T cells, etc.. The important piece of information is the contribution of tumor cell STAT3 to suppression of immune defense. We also provide evidence of the heterogeneity of tumor cells. The tumorigenic cells can be distinguished by gene expression signatures, and they segregate into distinct clusters with RAS dependency, cell cycle activity, and immune modulation. This result indicates that successful therapeutics will need to impact all three distinct tumor cell clusters: RAS, cell cycle, and immune stimulation.

Referee #3:

Overall, this is well-executed work that convincingly confirms current understanding that STAT3 contributes to tumor growth and behaviors in PDAC. What is missing, however, is further clarification of how STAT3 contributes. The consequences caused by the approach chosen here, namely complete removal of STAT3, can be interpreted in several ways. (i) STAT3 is required as a protein, possibly in a scaffolding role; (ii) alternatively, its activation by Tyr-phosphorylation is necessary and hence transcription factor function; or (iii) what is observed are compensatory effects caused by another protein that fills the void created by the removal of STAT3.

An answer to this question would go beyond the current understanding, and an attempt should be made to provide further insights. This seems pertinent in light of the reported results, such as the substantial transcriptional deregulation caused by STAT3 deletion shown in Fig. 2.

Also, in Fig 2, the re-expression of STAT3 partially restored tumorigenicity in nude mice, but if that can be achieved with a non-activatable STAT3 mutant has not been assessed.

The Reviewer raises an important question. We did evaluate the ability of a mutant STAT3 allele that replaces tyrosine 705 with a phenylalanine (STAT3^{Y705F}) to reconstitute tumorigenicity to the KRAS/STAT3 DKO cells. We have now added the results to Figure 2F, EV2F, and the text. (pg. 8) The frequency of tumor formation with STAT3^{Y705F} was only 1/8 and this tumor formed with

increased latency. The reduction in tumor formation compared with wt STAT3 indicates that activation of STAT3 by tyrosine phosphorylation is a major contributor to its function in tumorigenicity. Others reported transcriptional activity of unphosphorylated STAT3 (Philips et al. 2022), and this may have contributed to the tumor that was detected.

Since the loss of STAT3 with loss of KRAS eliminates tumorigenicity, the proposition of a functional compensatory protein is unclear.

In Fig 3 it is shown that KRAS inhibition leads to PDAC tumor regression. It remains unclear if the massive increase in STAT3 activation that accompanies this process contributes to tumorigenesis at all, and if yes, in pro- or anti-tumorigenic capacity.

We performed immunohistochemistry staining of tyrosine phosphorylated STAT3 in KPC tumors before and after mice were treated with the KRAS inhibitory regiment only to evaluate a potential increase in function (Figure 3). We agree that the increase in activation of STAT3 in response to these drugs does not necessarily indicate a role in regression. For that reason, we performed a regression study comparing tumors formed by the parental KPC EV cells and the STAT3 KO cells. KPC tumors that express STAT3 were more resistant to KRAS pathway inhibition, whereas the STAT3 KO tumors regressed more readily within a one-week period.

Dear Dr. Reich,

Thank you for the submission of your revised manuscript to our editorial offices. I have now received the reports from two of the three referees that I asked to re-evaluate the study, you will find below. Referee #2 declined my invitation to re-assess the manuscript and was unresponsive to my further messages to convince her/him otherwise. However, going through your point-by-point-response, I consider the concerns of referee #2 as adequately addressed. As you will see, the other two referees now support the publication of your study in EMBO reports.

Before we can proceed with formal acceptance, I have these editorial requests I ask you to address in a final revised manuscript:

- As your manuscript has only 4 main and 5 EV figures we plan to publish your manuscript as Report. For a Scientific Report we require that results and discussion sections are combined in a single chapter called "Results & Discussion". Please do this for your manuscript. For more details please refer to our guide to authors:
<http://www.embopress.org/page/journal/14693178/authorguide#researcharticleguide>

- Please re-arrange the figures to have not more than 5 final main and 5 final EV figures. Please move the information shown in the single Appendix figure to the EV figures and remove the Appendix file.

Moreover, figures should have only one page. Thus please arrange Figure EV2 differently or rename the panels in order to have 5 final one-page EV figures. Please also make sure that the final EV figures are named as Figure EVx (also in their legends) and called out like this. Finally, please check that all callouts are changed appropriately, and that each figure panel of the final main and EV figures is called out separately and sequentially.

- We now use CRediT to specify the contributions of each author in the journal submission system. CRediT replaces the author contribution section. Please use the free text box to provide more detailed descriptions and do not provide your final manuscript text file with an author contributions section. See also our guide to authors:
<https://www.embopress.org/page/journal/14693178/authorguide#authorshippinguidelines>

- We updated our journal's competing interests policy in January 2022 and request authors to consider both actual and perceived competing interests. Please review the policy <https://www.embopress.org/competing-interests> and update your competing interests if necessary. Please name this section 'Disclosure and Competing Interests Statement' and put it after the Acknowledgements section.

- Please order the manuscript sections like this, using these names:

Title page - Abstract - Keywords - Introduction - Results & Discussion - Methods - Data availability section - Acknowledgements - Disclosure and Competing Interests Statement - References - Figure legends - Expanded View Figure legends

- Please remove the mention of published datasets from the data availability section (DAS). This section is restricted to deposited datasets produced during the study. Please add these as data citations to the reference list and use appropriate callouts. See the section 'data citation' here:

<https://www.embopress.org/page/journal/14693178/authorguide#referencesformat>

- Please also remove this sentence from the DAS: 'Additional information and/or reagents are available from the authors on request.' Moreover, please add direct URL for the dataset GSE275858 and make sure this is public latest upon online publication of the study.

- Please add scale bars of similar style and thickness to microscopic images, using clearly visible black or white bars (depending on the background). Please place these in the lower right corner of the images themselves. Please do not write on or near the bars in the image but define the size in the respective figure legend. Presently, some scale bars are too small or have text nearby or in/on them. Please check.

- Please check again that the number "n" for how many independent experiments were performed, their nature (biological versus technical replicates), the bars and error bars (e.g. SEM, SD) and the test used to calculate p-values is indicated in the respective figure legends. Please also check that all the p-values are explained in the legend, and that these fit to those shown in the figure. Please provide statistical testing where applicable. Please avoid the phrase 'independent experiment' but clearly state if these were biological or technical replicates. Please also indicate (e.g. with n.s.) if testing was performed, but the differences are not significant. In case n=2, please show the data as separate datapoints without error bars and statistics. See also:

<http://www.embopress.org/page/journal/14693178/authorguide#statisticalanalysis>

If $n < 5$, please show single datapoints for diagrams. Presently, some diagrams show only partial statistics or the 'n.s.' seems missing. Moreover, please do not just put a p-value somewhere above the bars, but indicate by a line which datasets have been compared (as done in Fig. 3B). Please check. Moreover:

- Please note that the box plots need to be defined in terms of minima, maxima, centre, bounds of box and whiskers, and percentile in the legends of figures 1F, 2B, D, 4C, EV1 C.

- Please note that information related to n is missing in the legends of figures 1F, 2B, D; EV1 E, EV2 F, H.

- Please note that $n=2$ in figure 4B (see above).

- Please note that the error bars are not defined in the legends of figures 1H, 3B, 4B, EV1 E, G.

- Please note that the exact p values are not provided in the legends of figures 4B, C.

- Please indicate the statistical test used for data analysis in the legends of figures 1F, 2B, D; 3B; 4B, C; EV2 F.

- Please add to each legend (main, and EV figures, where applicable) a 'Data Information' section (or name the provided section like this) explaining the statistics used or providing information regarding replicates and scales. See:

- Thank you for providing the requested source data (SD). Please upload this as one folder per main figure (with all files for one figure in one folder and ZIPed together).

- During our figure integrity check, we noted that there are panel reuses between Fig. 2E and Fig. EV2D (HNF4A, ERK1/2) and within Figs. EV2A, EV2C and EV2D (RASG12D, ERK1/2). If these are intentional, please clearly indicate the reuse of the panels in the respective legends.

- Please confirm that for all Western blot panels (main and EV figures) the loading control was run on the same gel as the other proteins detected. Please note that we discourage comparisons between samples on different gels/blots, even if the samples derive from one experiment, as confounding factors reduce comparability. If unavoidable, the figure legend must state that the samples derive from the same experiment and that gels/blots were processed in parallel. If a 'representative' loading control is shown for multiple gels/blots, the intra-gel controls should be shown in the source data files and the figure legends should describe the data displayed accurately. See our author guidelines:

<https://www.embopress.org/page/journal/14693178/authorguide#datapresentationformat> (section 'Electrophoretic gels and blots').

and

<https://www.embopress.org/image-integrity>

Thus, please also provide the source data files (uncropped images) for the Western blots shown in the EV figures. Please upload these as one ZIPed folder containing all the source data for the EV figures in separate folders.

In addition, I would need from you uploaded separately:

- a short, two-sentence summary of the manuscript (not more than 35 words).

- two to four short (!) bullet points highlighting the key findings of your study (two lines each).

- a schematic summary figure as separate file that provides a sketch of the major findings (not a data image) in jpeg or tiff format (with the exact width of 550 pixels and a height of not more than 400 pixels) that can be used as a visual synopsis on our website.

Please use this link to submit your revision: <https://embor.msubmit.net/cgi-bin/main.plex>

Best,

Referee #1:

I have no more comments.

Referee #3:

D'Amico and colleagues answer my questions to my satisfaction. With the new data of Stat3-Y705F this manuscript shows the role Stat3 activation in tumor growth very well. In my opinion this manuscript is a great contribution to the understanding of the requirement of Stat3 in KRAS-depleted conditions.

All editorial and formatting issues were resolved by the authors.

Nancy Reich
Stony Brook University
Department of Microbiology and Immunology
Nichols Road
Stony Brook, NY 11794
United States

Dear Dr. Reich,

I am very pleased to accept your manuscript for publication in the next available issue of EMBO reports. Thank you for your contribution to our journal.

Yours sincerely,
